# *Caenorhabditis elegans* LET-413 Scribble is essential in the epidermis for growth, viability, and directional outgrowth of epithelial seam cells

**Amalia Riga, Janine Cravo, Ruben Schmidt, Helena R. Pires, Victoria G. Castiglioni, Sander van den Heuvel, Mike Boxem** *

Division of Developmental Biology, Institute of Biodynamics and Biocomplexity, Department of Biology, Faculty of Science, Utrecht University, Utrecht, The Netherlands

* m.boxem@uu.nl

**Data Availability Statement:** All relevant data are within the manuscript and its Supporting Information files.

## Abstract

The conserved adapter protein Scribble (Scrib) plays essential roles in a variety of cellular processes, including polarity establishment, proliferation, and directed cell migration. While the mechanisms through which Scrib promotes epithelial polarity are beginning to be unraveled, its roles in other cellular processes including cell migration remain enigmatic. In *C. elegans*, the Scrib ortholog LET-413 is essential for apical–basal polarization and junction formation in embryonic epithelia. However, whether LET-413 is required for postembryonic development or plays a role in migratory events is not known. Here, we use inducible protein degradation to investigate the functioning of LET-413 in larval epithelia. We find that LET-413 is essential in the epidermal epithelium for growth, viability, and junction maintenance. In addition, we identify a novel role for LET-413 in the polarized outgrowth of the epidermal seam cells. These stem cell-like epithelial cells extend anterior and posterior directed apical protrusions in each larval stage to reconnect to their neighbors. We show that the role of LET-413 in seam cell outgrowth is likely mediated largely by the junctional component DLG-1 discs large, which we demonstrate is also essential for directed outgrowth of the seam cells. Our data uncover multiple essential functions for LET-413 in larval development and show that the polarized outgrowth of the epithelial seam cells is controlled by LET-413 Scribble and DLG-1 Discs large.

## Author summary

Most cells in multicellular organisms are organized along a directional axis of cell polarity. One protein that is important for this polarized organization is the conserved polarity regulator Scribble. This protein has several functions, including forming the basolateral domains of cells, promoting the formation of cell junctions, and promoting cell migration. How Scribble performs these functions is not fully understood. In this paper we study the role of Scribble during larval development of the small nematode *Caenorhabditis elegans*

**Funding:** This work was supported by the Dutch Research Council (NWO)-ALW Open Program 824.14.021 and NWO-VICI 016.VICI.170.165 grants to M. B., and the European Union's Horizon 2020 research and innovation programme under the Marie Skłodowska-Curie grant agreement No. 675407 – PolarNet to M.B. and S.v.d.H. The funders had no role in study design, data collection and analysis, decision to publish, or preparation of the manuscript.

**Competing interests:** The authors have declared that no competing interests exist.

using an inducible protein degradation system. We show that Scribble, called LET-413 in *C. elegans*, is essential in the epidermal epithelium for animal development, as depletion of LET-413 in only this tissue blocks growth. We also demonstrate that LET-413 is required for the polarized outgrowth of an epithelial cell type called the seam cells, a process resembling cell migration. Finally, we show that one major function of LET-413 in seam cell outgrowth is the localization of the junctional component Discs large (DLG-1), which we demonstrate is also essential for this process. Our data thus uncover multiple essential functions for LET-413 in larval development and provide new insights into how the directional outgrowth of epithelial seam cells is controlled.

## Introduction

Epithelial cells establish molecularly and functionally distinct apical, basal, and lateral membrane domains to function as selectively permeable barriers. Epithelial cell polarity is established through mutually antagonistic interactions between conserved groups of cortical polarity regulators, including the Par, Crumbs and Scribble modules [1–4]. The Scribble polarity module, which consists of the proteins Scribble (Scrib), discs large (Dlg), and lethal giant larvae (Lgl), plays conserved roles in the establishment of basolateral identity and formation of cell junctions [5–13]. In addition, Scribble module proteins are involved in the regulation of cell proliferation and migration. In *Drosophila*, *scrib*, *dlg*, and *lgl* function as suppressors of neoplastic overgrowth of imaginal disks [14–17]. Many human tumors show altered Scrib protein levels or protein mislocalization, and in both *Drosophila* and mammalian tumor models, loss of Scrib increases the malignant and metastatic potential of oncogenic stimuli such as activation of Ras, Raf, Notch or Akt [18–21].

The ability of Scrib to affect metastasis may be linked to its role as a regulator of cell migration. In *Drosophila*, *scrib* is required for migration of epithelial cells during dorsal closure [2]. In vertebrates, Scrib is involved in the migration of multiple cell types [7,22–27]. How Scrib affects cell migration is not well understood and may differ between cell types. In several epithelial cell types, Scrib is thought to regulate actin dynamics at the leading edge by promoting the recruitment or activation of the Rho-family GTPases Rac and Cdc42, or their effector proteins p21-activated kinases (PAKs) [7,24,28]. In other cell types, effects on cell migration do not appear to be mediated by small GTPases. In endothelial cells, Scrib regulates directed cell migration on fibronectin-coated surfaces by binding to and protecting surface integrin α5 from lysosomal degradation [23]. In MDCK cells, Scrib affects cell migration through loss of E-cadherin-mediated cell adhesion [26]. Finally, in dendritic cells and several cancer cell lines, Scrib was found to control cell migration downstream of the transmembrane semaphorin 4A (Sema4A) [27]. In this context, cell migration appeared to be promoted by the downregulation of the activities of Scrib, Cdc42, and Rac1 [27]. These examples illustrate the complexities of Scrib in cell migration.

The *C. elegans* genome encodes a single Scribble protein termed LET-413 that is essential for junction formation and epithelial polarization. In epithelia of embryos lacking *let-413* activity, junctional proteins fail to assemble into a continuous subapical belt and are found expanded through the lateral domain [5,10,11,29–31]. In addition, *let-413* embryonic epithelia show basolateral invasion of apical proteins including PAR-3, PAR-6, and the intermediate filament protein IFB-2 [11,29]. Ultimately, *let-413* embryos arrest due to a failure to elongate beyond the 1.5–2-fold stage. Investigating *let-413* in later stages of *C. elegans* development could provide further insights in the cellular pathways in which *let-413* participates, but the

roles of *let-413* in larval development are not well characterized. Larval *let-413(RNAi)* causes sterility due to dysfunction of the spermathecal epithelium, where *let-413* was shown to be required for assembly of apical junctions and the maintenance of basolateral identify [32]. However, no defects in growth rate or motility were reported. The role of *let-413* was also investigated in the intestine, using an intestine-specific CRISPR/Cas9 somatic mutant [33]. Using this approach, LET-413 was shown to promote endocytic recycling. However, intestinal *let-413* CRISPR mutants continue larval development. Whether LET-413 is important in other larval tissues remains unclear. Moreover, it is not known if the role of Scrib in cell proliferation and migration is conserved in *C. elegans*.

Here, we use inducible protein degradation to investigate the roles of LET-413 in postembryonic epithelial tissues of *C. elegans*. Consistent with previous data, the presence of LET-413 in the intestine in larval stages was not essential for larval development. In contrast, we find that LET-413 is essential in the epidermis to support larval growth and viability, junctional integrity, and the directional outgrowth of the epithelial seam cells. The stem cell-like seam cells undergo an asymmetric division during each larval stage, followed by fusion of the differentiating anterior daughter cells with the surrounding epidermal syncytium. The remaining seam cells then form anterior–posterior directed protrusions of the apical domain to re-establish cell–cell contacts. To date, the mechanisms that mediate this dynamic shape change are poorly understood. To begin to understand the roles LET-413 may play in this process, we performed time-lapse imaging of membrane dynamics and investigated the localization of actin during seam cell outgrowth. The seam cells showed active membrane dynamics with enrichment of actin in the protrusions, indicating that seam cell outgrowth is an active actin-driven process. This raised the possibility that LET-413 regulates actin dynamics through small GTPases. However, inactivation of Rac family members or CDC-42 only resulted in a partial block of seam cell outgrowth. Because LET-413 depletion resulted in disruptions in the integrity of cell junctions, we next investigated the roles of the DLG-1/AJM-1 and cadherin-catenin junctional complexes in protrusion formation. We show that LET-413 is required for proper junctional localization of HMR-1 and DLG-1, and that DLG-1, but not HMR-1, is essential for seam cell outgrowth. Thus, LET-413 controls the directed extension of seam cells in large part by promoting the assembly or stability of DLG-1 at the apical junctions. Nevertheless, seam cells depleted of DLG-1 show some remaining membrane dynamics, indicating that LET-413 may have additional functions in seam cell outgrowth. Together, our data show that the role of Scrib in promoting protrusive cell shape changes is conserved in *C. elegans* and demonstrate essential roles for a LET-413 Scrib/DLG-1 Discs large pathway in larval development and directed seam cell outgrowth.

## Results

### LET-413 is essential for larval growth and viability

To investigate the role of LET-413 in larval epithelia of *C. elegans*, we used the auxin inducible degradation (AID) system, which enables degradation of AID-degron-tagged proteins in a time- and tissue-specific manner [34,35]. We tagged endogenous LET-413 at the N-terminus, shared by all predicted isoforms, with the AID degron and a green fluorescent protein (GFP) (Fig 1A). Before morphogenesis, we detected LET-413 ubiquitously at cell membranes by spinning-disc fluorescence microscopy (Fig 1B). From the bean stage onward, LET-413 localized to the basolateral membrane domain and at intercellular junctions of epidermal and intestinal cells (Fig 1B). In larval stages and adults, LET-413 was expressed in the intestine, epidermis, excretory canal, and the reproductive system (vulva, uterus, and spermatheca), where it also appeared to localize to the basolateral membranes and cell junctions (Figs 1C and S1). These

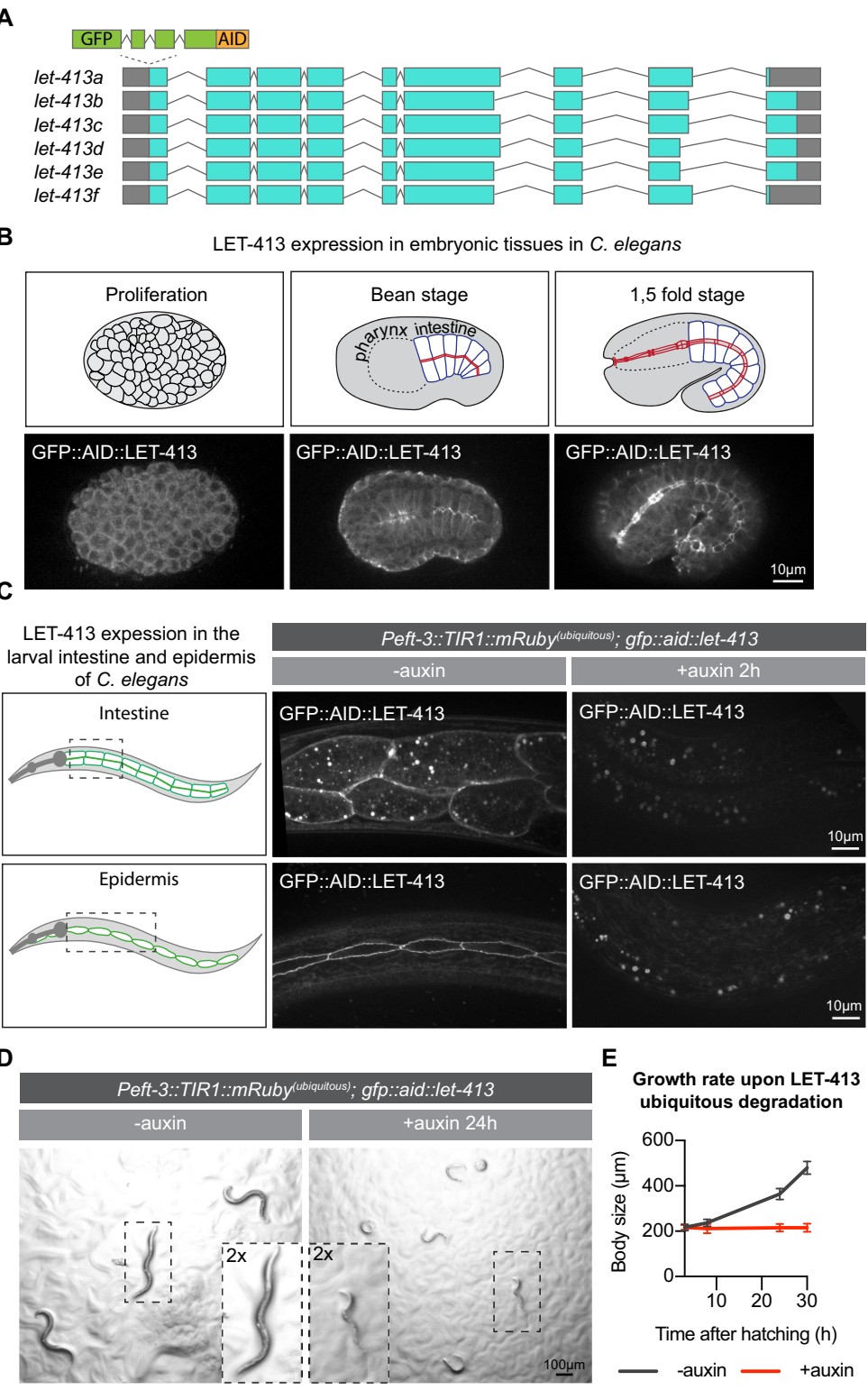

**Fig 1. LET-413 is essential for larval development.** (**A**) Schematic representation of predicted *let-413* splice variants and the insertion site of sequences encoding a green fluorescent protein (GFP) and the auxin-inducible degradation degron (AID). (**B**) Expression of LET-413 in embryonic development (strain BOX466). (**C**) Expression of LET-413 in the intestine and epidermis of *Peft-3::TIR1::mRuby; GFP::AID::let-413* L4 animals in the absence (-auxin) or presence (+auxin) of auxin for 2 h (strain BOX469). Drawings are a schematic representation of the LET-413 expression pattern.

Images are maximum intensity projections. In this and all other figures, 3 mM auxin is used (**D, E**) Growth of *Peft-3::TIR1::mRuby; GFP::AID::let-413* animals in the absence (-auxin) or presence (+auxin) of auxin (strain BOX469). Images in D were acquired 24 h post hatching. Growth curves in E show mean length ± SD upon continuous exposure to auxin. N = 8, 10, 12, and 9 for -auxin; and 10, 14, 19, and 13 for +auxin.

localization patterns are consistent with the results from previous studies that used antibody staining and transgene expression [10,11,32,36].

To determine whether LET-413 is essential for larval development, we degraded LET-413 using ubiquitously expressed TIR1 under control of the *eft-3* promoter, which is active in most or all tissues during larval development (Zhang et al., 2015). After 2 h of exposure of L1 larvae to auxin, LET-413 levels were depleted throughout the animal body. Quantifications of the GFP levels at epidermal cell junctions and the basolateral domain of intestinal cells revealed loss of GFP enrichment after auxin addition, confirming efficient degradation of LET-413 (Fig 1C). Degradation of LET-413 from hatching onward resulted in a growth arrest and larval lethality (Fig 1D and 1E). At 6 h of development, LET-413-depleted animals were already measurably smaller than control animals not treated with auxin (Fig 1E). At 24 h after hatching, we observed not only a lack of growth, but also ~80% larval lethality, as evidenced by lack of response to physical stimulation and lack of motility. These results show that LET-413 is essential for larval development.

## LET-413 is essential in the larval epidermis, but not the intestine

We next wanted to determine which larval tissue(s) contribute to the observed growth defect and lethality. We focused on two major larval epithelial tissues: the intestine and the epidermis. To deplete LET-413 specifically in these tissues, we made use of single-copy integrated lines expressing TIR1 from the tissue-specific promoters *Pelt-2* and *Pwrt-2*, which are active in the intestine and epidermis, respectively [37]. We examined intestinal morphology using an endogenous fusion of mCherry to the junctional protein DLG-1 Discs large but did not observe any defects in the characteristic junctional localization pattern, despite a lack of detectable LET-413 (Fig 2A and 2B and 2C). Animals depleted of LET-413 in the intestine also grew as normal (Fig 2D). These data indicate that intestinal functioning does not critically depend on the continuous presence of LET-413. We next tested the requirement for LET-413 in the epidermis. Degradation of LET-413 occurred rapidly, with no GFP signal detected in the junctions of seam cells 2 h after the addition of auxin (Fig 2E and 2G). In contrast to the intestine, degradation of LET-413 in the epidermis resulted in severe growth defects and larval lethality (Fig 2D and 2F). Compared to ubiquitous degradation of LET-413, the growth defect was slightly less severe and more variable, while the larval lethality at 24 h of development was similar (Fig 2H). These data show that LET-413 is essential for the functioning of the larval epidermis in *C. elegans*.

## LET-413 is required for seam cell extension and reattachment

The epidermis of *C. elegans* larvae largely consists of the syncytial hypodermal cell hyp7 and two lateral rows of end-to-end attached epithelial seam cells embedded within hyp7 (Fig 3A). Animals hatch with a complement of 10 seam cells on each side (H0–H2, V1–V6 and T) (Fig 3A), which undergo a reproducible pattern of asymmetric and symmetric divisions at specific times in development (Fig 3B) [38]. In each larval stage, the V1–V4 and V6 seam cells undergo an asymmetric division, which generates an anterior daughter that differentiates and fuses with the hypodermis, while the posterior daughter retains the seam cell fate. In the second larval stage, the asymmetric division is preceded by a symmetric division that generates two seam

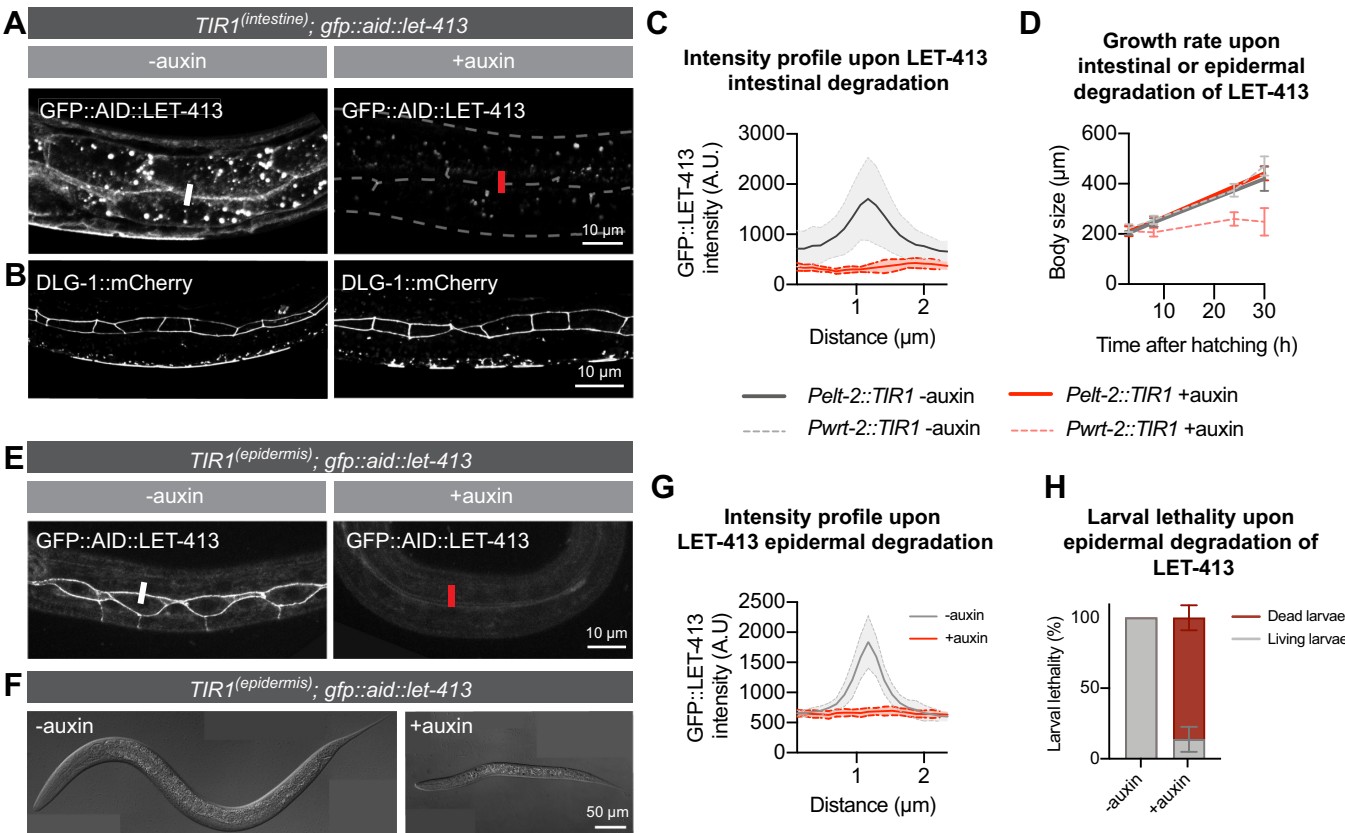

**Fig 2. LET-413 is essential in the larval epidermis, but not the intestine.** (**A**) Distribution of LET-413 in the intestines of *Pelt-2::TIR1::BFP; GFP::AID::let-413; dlg-1::mCherry* animals 36 h post hatching in the absence (-auxin) or presence (+auxin) (strain BOX530). Images are a single focal plane at the level of junctions between opposing intestinal cells. White and red bars are an example of sites used for fluorescence intensity quantification in C. (**B**) Distribution of DLG-1::mCherry in the same animals as in A. Images are maximum intensity projections of the intestinal cell junctions. (**C**) Intensity profiles of GFP::LET-413 fluorescence at junctions of opposing intestinal cells in *Pelt-2::TIR1::BFP; GFP::AID::let-413; dlg-1::mCherry* animals in the absence (-auxin) or presence (+auxin) of auxin for 30 h (strain BOX530). Solid lines and shading represent mean ± SD. N = 6 animals for -auxin and 4 animals for +auxin. (**D**) Growth curves of *Pelt-2::TIR1::BFP; GFP::AID::let-413* and *Pwrt-2::TIR1::BFP; GFP::AID::let-413* animals in the absence (-auxin) or presence (+auxin) of auxin (strains BOX449 and BOX530). Solid lines represent animals that express TIR1 in the intestine (*Pelt2::TIR1::BFP*) and dashed lines in the epidermis (*Pwrt2::TIR1::BFP*). Data show mean ± SD. N = 8, 10, and 10 animals for *Pelt2::TIR1* -auxin; 13, 13, and 10 for *Pelt2::TIR1* +auxin; 14, 14, 13 and 10 for *Pwrt2::TIR1* -auxin; and 14, 16, 28, and 11 for *Pwrt2::TIR1* +auxin. (**E**) Distribution of LET-413 in the epidermis of *Pwrt-2::TIR1::BFP; GFP::AID::let-413* L1 stage animals in the absence (-auxin) or presence (+auxin) of auxin for 2 h (strain BOX449). White and red bars are an example of sites used for fluorescence intensity quantification in G. (**F**) Example of growth of *Pwrt-2::TIR1::BFP; GFP::AID::let-413* animals in the absence (-auxin) or presence (+auxin) of auxin 24 h post hatching. (**G**) Intensity profiles of GFP::LET-413 fluorescence at seam–hyp7 junctions in *Pwrt-2::TIR1::BFP; GFP::AID::let-413* animals in the absence (-auxin) or presence (+auxin) of auxin for 2 h (strain BOX449). Solid lines and shading represent mean ± SD. N = 5 animals for -auxin and 4 for +auxin. (**H**) Percentage of larval lethality upon degradation of epidermal LET-413 in *Pwrt-2::TIR1::BFP; GFP::AID::let-413* larvae grown in the presence of auxin from hatching (strain BOX449). Data show mean ± SD.

cells. V5 follows a similar division pattern, except for the anterior daughter of the L2 division which becomes a neuroblast that generates a sensory structure termed the posterior deirid sensillium. Following the asymmetric divisions and fusions of the anterior daughters, the remaining seam cells extend anterior and posterior protrusions towards their neighbors and reattach, closing the gaps left by the fused cells.

To understand how loss of LET-413 affects the epidermal epithelium, we followed the seam cell division pattern in animals treated with auxin from hatching, using seam cell-specific GFP reporters that mark the cell membrane (*Pwrt-2::GFP::PH*$^{PLC1\delta}$) and DNA (*Pwrt-2::GFP::H2B*) [39]. Degradation of LET-413 did not affect the L1 asymmetric seam cell divisions or the fusion of the anterior daughters with hyp7 (Fig 3D). Following cell fusion, however, the

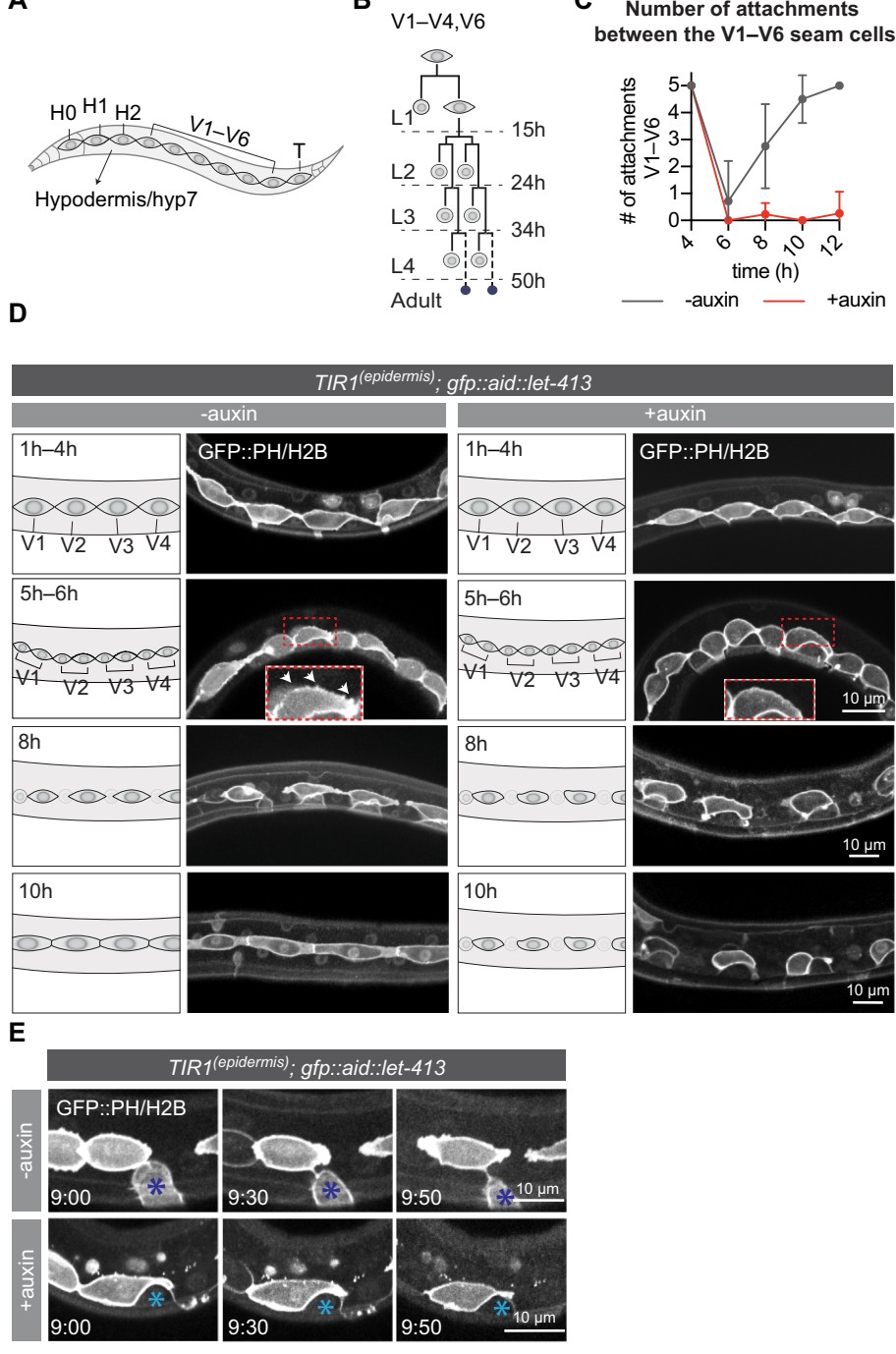

**Fig 3. Loss of LET-413 leads to failure in seam cell outgrowth and reattachment after the asymmetric divisions.**
(**A**) Schematic representation of the *C. elegans* epidermis (lateral view). (**B**) Division pattern of the V1–V4 and V6 seam cells. Dotted lines demarcate the larval stages. Asymmetric divisions generate one differentiating daughter cell (round anterior cell) and one seam cell. During the L4 to adult molt, the remaining seam cells fuse and form a syncytium (small blue cells). (**C**) Number of attachments between the V1–V6 seam cells in LET-413-depleted (+auxin) or control animals (-auxin) (strain BOX582). Solid lines represent mean ± SD. N = 7, 7, 16, 16, and 11 for -auxin time points, and 7, 9, 15, 15, and 13 for +auxin time points. Genotype is *Pwrt-2::TIR1::BFP; GFP::AID::let-413; heIs63 [Pwrt-2::GFP::PH; Pwrt-2::GFP::H2B; Plin-48::mCherry]*. Time indicates hours post hatching. (**D**) Time series of L1 seam cell divisions and subsequent extension in LET-413-depleted (+auxin) or control animals (-auxin) (strain BOX582). Seam-specific GFP::H2B and GFP::PH mark DNA and cell membrane, respectively. Images are single apical focal planes. Times indicate hours post hatching. White arrows at 5–6 h time point indicate membrane protrusions. (**E**) P cell

retraction in LET-413-depleted (+auxin) or control animals (-auxin) (strain BOX582). Images were taken at 9 h post hatching +0 min, +30 min, and +50 min. Blue asterisks indicate P cells. Seam-specific GFP::H2B and GFP::PH mark DNA and cell membrane, respectively.

remaining seam cells failed to extend protrusions towards their neighbors and remained isolated (Fig 3C and 3D). In control animals, immediately following cell division the seam cells formed small filopodia-like protrusions around the cell body (Fig 3D, 5h–6h, white arrows). The apical domains of the seam cells then formed larger lamellipodium-like extensions directed towards the adjacent seam cells, and cells reattached around 10 h after hatching (Fig 3C). Upon epidermal degradation of LET-413, the seam cells failed to form protrusions directed towards neighboring cells and remained unattached (Fig 3C and 3D). Depletion of LET-413 at later stages of development also resulted in isolated seam cells (S2 Fig). When depleting LET-413 just before the L2 divisions, we observed isolated clusters of >2 seam cells at a time where in control animals the posterior daughters had already reattached. This indicates that anterior seam cell daughters delay fusion or fail to fuse with hyp7, a defect not observed following L1 or L3 divisions. Presumably, the difference between larval stages is related to the unique division pattern of the seam cells in the L2 stage, in which the asymmetric divisions are preceded by a symmetric division that doubles the seam cell number. Nevertheless, LET-413 appears to be required for seam cell outgrowth throughout larval development.

Besides the seam cell extension defects, we also noticed defects in the retraction of the ventrolateral P cells (P1–P12). At hatching, 6 pairs of P cells cover the ventrolateral surface adjacent to V1 through V6 [38,40,41]. Around the middle of the L1 stage, adjacent P cells separate from each other. Next, the P cells retract ventrally and interdigitate to form a single row of cells on the ventral side [40–43]. In the presence of auxin, P cell pairs still became detached from their anterior and posterior neighbors but did not retract ventrally and appeared to stay in contact with the seam cells (Fig 3E). This failure to retract may be accompanied by a change in cell fate or gene expression, as we noticed that LET-413 depletion results in reduced expression in P cells of the *Pwrt-2*-driven GFP::H2B and GFP::PH marker proteins (Fig 3E). Finally, we tested whether LET-413 is required for migration of the Q neuroblasts, which are born within the lateral rows of seam cells but migrate anteriorly and posteriorly during L1 development [44]. However, LET-413 is not expressed in the Q cell lineage, and we did not observe any defects in the migration of the Q cells or their descendants upon epidermal degradation of LET-413 (S3A and S3B Fig).

Taken together, we conclude that LET-413 is required in the epidermis for outgrowth of the seam cells and retraction of the P cells, but not for migration of the Q neuroblasts.

## Actin dynamics and role of Rho-family small GTPases in seam cell extension

To begin to understand how LET-413 might control seam cell extension, we first examined this process in more detail. Time lapse imaging showed that the seam cell extensions are highly dynamic, reminiscent of protrusions formed by migrating cells (S1 Video). As cell migration relies on the dynamic reorganization of the actin cytoskeleton [45–47], we determined the localization of actin during seam cell outgrowth. An actin marker consisting of the actin-binding domain of VAB-10 fused to mCherry (*Plin-26::vab-10[ABD]::mCherry*) [48] was enriched in the protrusions of extending seam cells (Fig 4A, dashed boxes). In LET-413-depleted animals, we did not observe polarized enrichment of actin (Fig 4A). To determine if branched actin organization is required for protrusion formation, we inactivated the WAVE (Wiskott-Aldrich syndrome protein family verprolin-homologous) complex component GEX-2 by

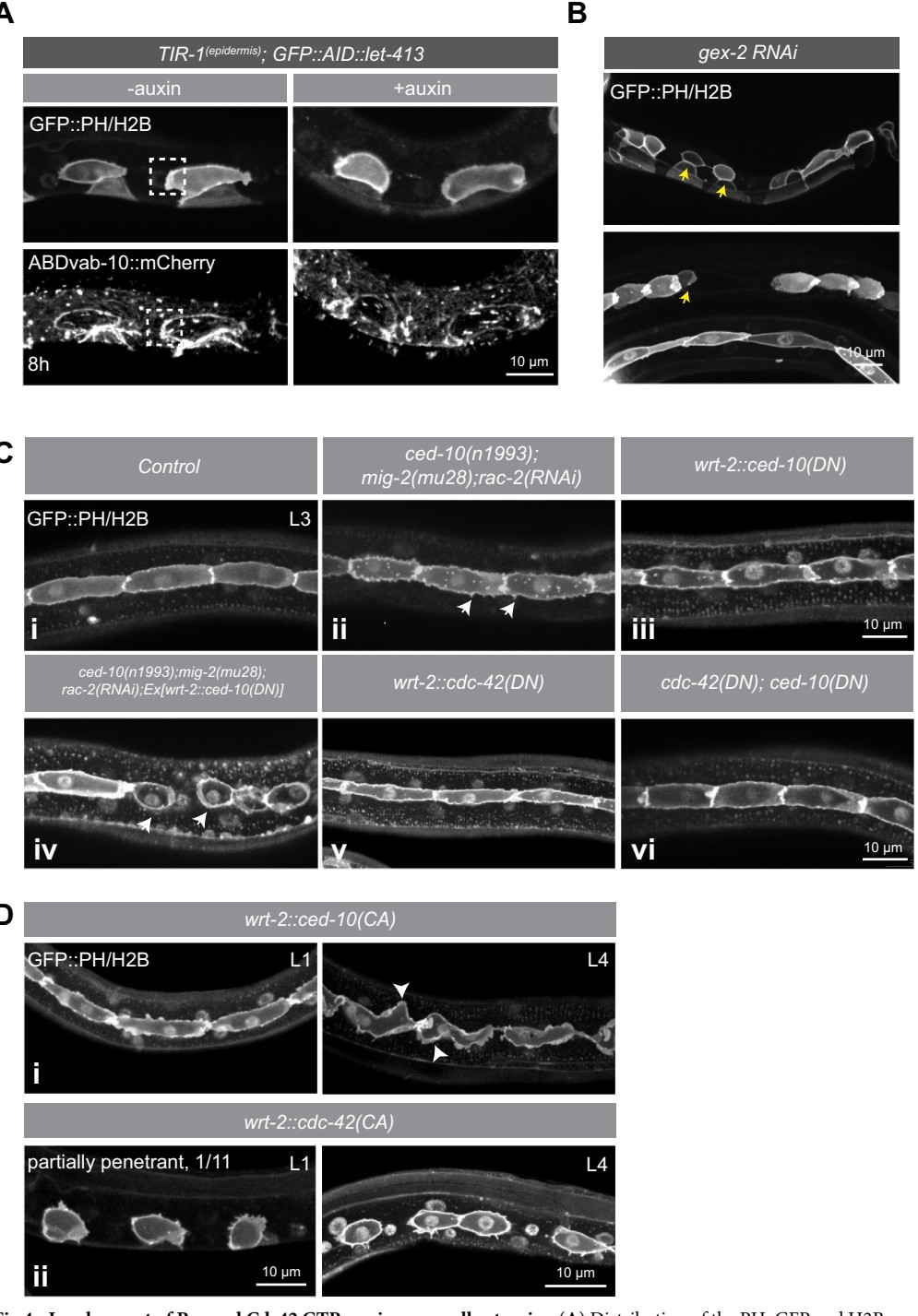

**Fig 4. Involvement of Rac and Cdc42 GTPases in seam cell extension** (**A**) Distribution of the PH::GFP and H2B:: GFP markers and the VAB-10[ABD]::mCherry actin probe in the epidermis of *Pwrt-2*::*TIR1*::*BFP; GFP::AID::let-413* animals without (-auxin) and in the presence of auxin (+auxin) at 8 h post hatching (strain BOX625). Images are single planes of the apical domain of the seam cells. White dotted boxes indicate the enrichment of VAB-10[ABD] in the seam cell protrusions. (**B**) Distribution of the GFP::PH and GFP::H2B marker in *gex-2(RNAi)* animals (strain SV1009). Yellow arrows point to defects in seam cell protrusions and seam cell shape. Defects were observed in 7/31 *gex-2 (RNAi)* escaper animals examined. Due to severity of phenotype, an exact larval stage cannot be determined. (**C**) Distribution of the GFP::PH and GFP::H2B markers in control animals, and in animals with indicated genetic backgrounds and methods of reducing the activity of Rac family members or CDC-42. i = control (strain SV1009), ii = triple Rac inactivation (strain BOX699), iii = CED-10(DN) (strain BOX697), iv = CED-10(DN) combined with

triple Rac inactivation strain, v = CDC-42(DN) (strain BOX747), and vi = combined expression of CED-10(DN) and CDC-42(DN) (SV1009 strain injected with CED-10(DN) and CDC-42(DN). White arrowheads in (ii) point to representative cortex abnormalities in the seam cells. Non-attached seam cell phenotype in (iv, arrowheads) was observed in 8/18 animals examined. (D) Distribution of the GFP::PH and GFP::H2B markers in L1 and L4 animals expressing constitutively active CED-10 (i) (strain BOX700) or CDC-42 (ii) (strain BOX739). Arrows in (i) indicate undirected ruffling. Unattached seam cells in (ii) were observed in 1/12 L1 animals and 30/34 L4 animals examined. Dominant negative and constitutively active variants of CED-10 and CDC-42 were expressed from extrachromosomal arrays and used mCherry or mTagBFP2 separated from CED-10 or CDC-42 by the F2A self-cleaving peptide to detect transgenic animals.

RNAi feeding. *gex-2* function is required for Arp2/3-mediated branched actin nucleation in migrating embryonic epidermal cells [49,50]. Because *gex-2(RNAi)* causes embryonic lethality, we placed adult animals on RNAi plates and examined rare escaper progeny. Animals also expressed VAB-10[ABD]::mCherry to discern RNAi-affected from non-RNAi-affected escapers. In escapers, we observed abnormal rounded seam cells and gaps between cells, indicative of a lack of seam cell extension (Fig 4B, yellow arrows). Together, these observations indicate that seam cell extension is a dynamic, actin-driven process.

One of the ways mammalian Scrib may promote cell migration is through regulating the activity or cortical localization of Rac or Cdc42 small GTPases, which in turn promote actin cytoskeleton rearrangements [7,25]. Indeed, inactivation of *C. elegans* CDC-42 by RNAi in larval stages was reported to result in defects in seam cell extension [51]. We therefore investigated the role of the Rac and Cdc42 families of GTPases in seam cell outgrowth. The *C. elegans* genome encodes 3 Rac-related proteins: CED-10 Rac, MIG-2 RhoG, and RAC-2 Rac [52]. Null alleles of *ced-10* are maternal effect embryonic lethal [50,52,53]. To be able to examine the function of Rac proteins in larval development, we therefore combined the hypomorphic *ced-10(n1993)* allele with the predicted *mig-2(mu28)* null allele, a combination that causes a strong reduction of protrusive activity in intercalating embryonic dorsal epidermal cells [54], and inactivated *rac-2* by RNAi. However, seam cell outgrowth and reattachment still occurred in *ced-10(n1993); rac-2(RNAi); mig-2(mu28)* animals (Fig 4Ci, ii). We did notice an increase in filopodia-like protrusions (arrowheads in Fig 4Cii), possibly due to a shift in the balance of activity between small GTPases. As an alternative approach to disrupt *ced-10* signaling, we expressed a dominant negative (DN) T17N mutant of CED-10 in the epidermis [54,55]. In animals expressing CED-10(DN), seam cells still developed protrusions and reattached (Fig 4Ciii). However, expressing CED-10(DN) in *ced-10(n1993); rac-2(RNAi); mig-2(mu28)* animals resulted in one or more detached seam cells in 8/18 animals examined (Fig 4Civ). Thus, strong disruption of Rac activity can interfere with the seam cell outgrowth process.

We next examined the contribution of CDC-42, the single Cdc42 subfamily member present in *C. elegans*. Like CED-10, *C. elegans* CDC-42 is essential for embryonic development. Hence, we expressed a dominant negative (DN) T17N mutant in the epidermis [54,55]. However, expression of CDC-42(DN) did not result in defects in the extension or reattachment of the seam cells (Fig 4Cv). Finally, we co-expressed the dominant negative CED-10 and CDC-42 constructs in the epidermis, but again did not observe defects in seam cell extension and reattachment (Fig 4Cvi). Thus, interfering with the functioning of these small GTPases was not sufficient to prevent the formation of directed seam cell protrusions to the same extent observed in LET-413 depleted animals, possibly because some Rac or Cdc42 activity remains present.

As an alternative approach to determine if Rac or Cdc42 signaling promotes protrusion formation, we expressed constitutively active (CA) Q61L mutants of CED-10 and CDC-42 in the epidermis [54,55]. Expression of CED-10(CA) caused severe seam cell morphology defects, characterized by the appearance of excessive membrane ruffles or filopodia around the cell

circumference (Fig 4Di). In L4 stage older animals, we also observed undirected protrusions of the seam cells (Fig 4Di, arrowheads). The expression of CDC-42(CA) resulted in rounded, detached seam cells (Fig 4Dii). The penetrance of the defects increased with age of the animals, ranging from 1 of 12 examined transgenic L1 animals to 30 of 34 L3 animals showing one or more gaps between the seam cells. Possibly this reflects accumulating CDC-42(CA) protein levels.

In summary, our data indicate that seam cell extensions are actin-driven, and that CED-10 and CDC-42 may play a role in reorganizing the actin cytoskeleton and driving anterior–posterior directed outgrowth. However, in contrast to LET-413 depletion, interfering with Rac or Cdc42 activity did not result in a complete block of seam cell outgrowth. Hence, our data do not support regulation of small GTPases as the primary mode of action of LET-413 in regulating seam cell outgrowth.

## Loss of LET-413 causes junction impairment but does not affect the localization of apical PAR-6 or basolateral LGL-1

As regulation of actin dynamics may not be the primary mechanism through which LET-413 controls seam cell extension, we next investigated the effects of LET-413 depletion on other aspects of the seam cells. In embryonic epithelial tissues, LET-413 is important for junction assembly or integrity as well as for the basolateral exclusion of apical polarity determinants [5,10,11]. To determine whether these roles are conserved in the larval epidermis, we examined the integrity of cell junctions and localization of apical–basal polarity markers upon depletion of LET-413. We first examined the distribution of the apical polarity regulator PAR-6 and the basolateral protein LGL-1. Under normal conditions, PAR-6 localizes to the apical surface of the seam cells, with enrichment at the cell junctions (Fig 5A and 5C) [37]. Depletion of LET-413 did not affect the levels of PAR-6 at the apical and junctional domains (Fig 5A and 5C), nor did it cause visible invasion of PAR-6 in the basolateral domain (Fig 5A). We also did not detect apical invasion or reduced basolateral levels of LGL-1 (Fig 5B and 5D). Thus, the depletion of LET-413 results in severe outgrowth defects but does not appear to affect the maintenance of apical–basal polarity.

We next investigated the integrity of the *C. elegans* apical junctions (CeAJs) using a strain that expresses endogenously tagged HMR-1::GFP E-cadherin and DLG-1::mCherry Discs large, to mark the cadherin-catenin complex (CCC) and DLG-1/AJM-1 complex (DAC), respectively. DLG-1 forms a homogeneous junctional band around the seam cells, while HMR-1 shows a more punctate pattern (Fig 5E and 5F). Epidermal depletion of LET-413 caused severe defects in the localization pattern of both HMR-1 and DLG-1. HMR-1 puncta became sparser, while DLG-1 was no longer localized in a continuous belt and instead localized to discontinuous short stretches and puncta (Fig 5E and 5F). HMR-1 localization was already highly abnormal at 4 h post-hatching, well before the first asymmetric division of the seam cells (Fig 5E). The first defects in DLG-1 localization were apparent at 5 h post-hatching and increased in severity over time (S4 Fig). Together, our results demonstrate that the loss of LET-413 causes junction impairment but does not affect the apicobasal polarization of the seam cells.

## LET-413 acts upstream of DLG-1 in the regulation of seam cell outgrowth

The severe defects in HMR-1 and DLG-1 localization we observed upon depletion of LET-413 raised the possibility that impairment of the cadherin-catenin or DLG-1/AJM-1 junctional complexes underly the outgrowth phenotype of LET-413-depleted seam cells. We therefore investigated the effects of tissue-specific depletion of HMR-1 and DLG-1 on seam cell extension. For HMR-1, we made use of an approach that combines auxin-dependent depletion with

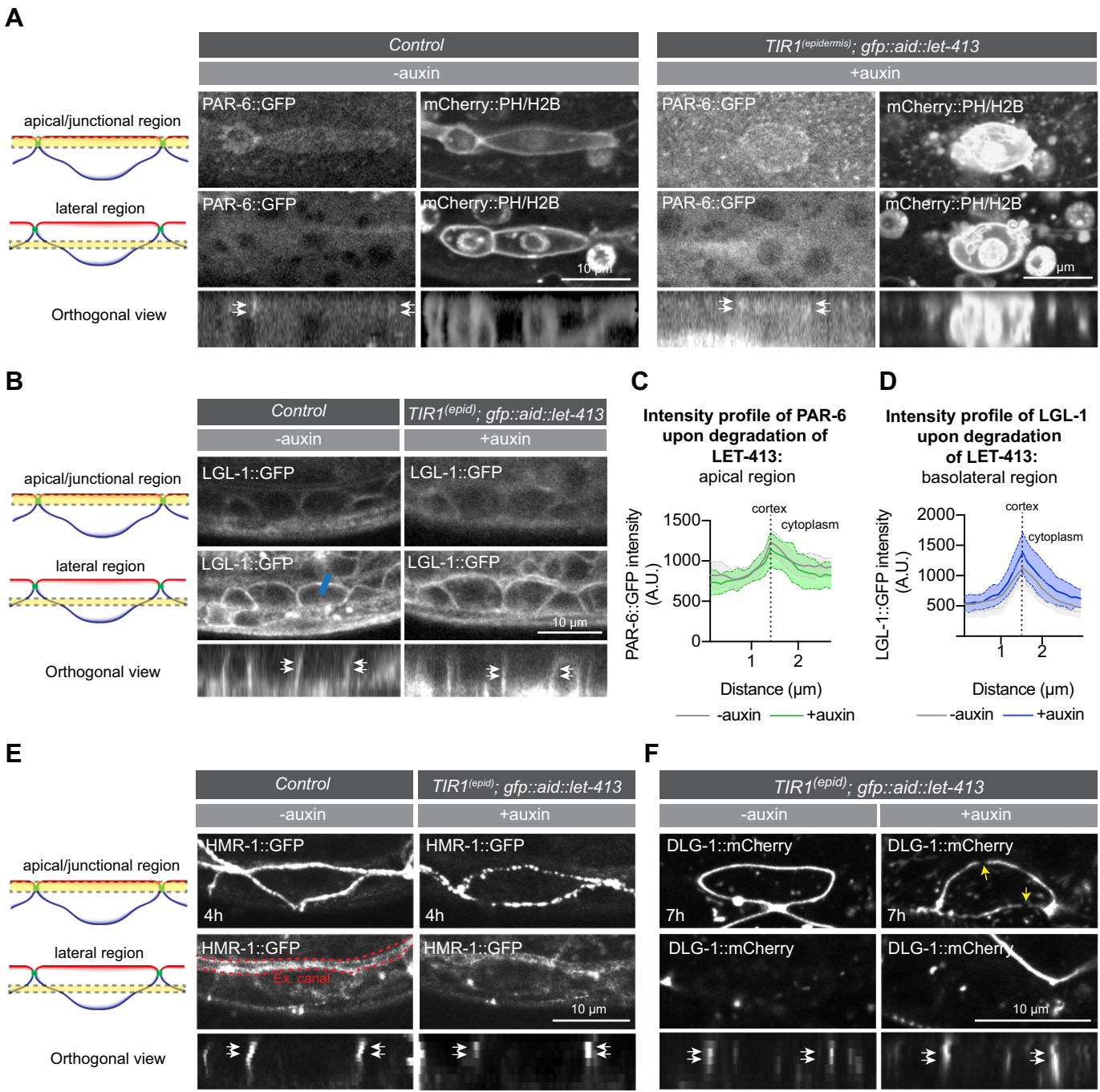

**Fig 5. Loss of LET-413 causes junction impairment but does not affect the localization of the polarity regulators PAR-6 and LGL-1.** (**A, C**) Distribution and quantification of PAR-6::GFP upon depletion of LET-413 at 5 h post hatching (strain BOX692). As both PAR-6 and LET-413 are fused to GFP and their localization overlaps, the control animals (strain BOX693) do not carry the *Pwrt-2::TIR1::BFP* and *GFP::AID::let-413* gene fusions. Graphs show mean apical GFP fluorescence intensity ± SD at the hyp7–seam cell junction (apical/junctional plane). N = 5 animals for both conditions. (**B, D**) Distribution and quantification of LGL-1::GFP in the epidermis of *lgl-1::GFP* animals without auxin (strain BOX041) and in *Pwrt-2::TIR1::BFP; GFP::AID::let-413; lgl-1::GFP* animals in the presence of auxin (strain BOX694) at 5 h post hatching. Graphs show the mean junctional GFP fluorescence intensity ± SD at the hyp7–seam cell junction (blue bar in D). N = 5 animals for both conditions. (**E**) Distribution of HMR-1 in the epidermis of *hmr-1::GFP* animals (strain SV1955) without auxin (-auxin) and *hmr-1::GFP; Pwrt-2::TIR1::BFP; GFP::AID::let-413* animals (strain BOX584) in the presence of auxin (+auxin) at 4 h post hatching. (**F**) Distribution of DLG-1::mCherry in the epidermis of *Pwrt-2::TIR1::BFP; GFP::AID::let-413; dlg-1::mCherry* animals without (-auxin) and in the presence of auxin (+auxin) at 7 h post hatching (strain BOX585). Yellow arrows indicate gaps in the normally continuous DLG-1::mCherry belt.

Cre-mediated knockout of the *hmr-1* locus (Fig 6A) (J. Cravo and S. van den Heuvel, manu-script in preparation). Using this approach, HMR-1 levels were reduced below the level of detectability by microscopy (Fig 6B and 6D). However, the seam cells still formed anterior and posterior protrusions (Fig 6C) and reattached to the neighboring cells. Thus, HMR-1 is dis-pensable for protrusion formation and the outgrowth of the seam cells.

To deplete DLG-1, we generated an endogenous C-terminal fusion of DLG-1 with AID:: GFP. DLG-1 proved relatively resistant to auxin-mediated depletion. We therefore extended the auxin treatment by exposing newly hatched larvae to auxin for 24 h before initiating devel-opment. Additionally, we used an alternative degron sequence (miniIAA7) that results in improved degradation efficiency in mammalian cells [56] (Fig 6E). Using this approach, DLG-1 levels were reduced to ~6% of non-auxin-treated controls (Fig 6F and 6H). The seam cells in DLG-1-depleted animals showed severe outgrowth defects and did not extend towards the neighboring seam cells (Fig 6G). Importantly, depletion of DLG-1 did not affect the localiza-tion or levels of LET-413 in the epidermis (Fig 6I and 6J). This indicates that the seam cell out-growth defects in LET-413-depleted larvae are a consequence of the defects in DLG-1 organization. Thus, in agreement with previous observations in the embryo and spermatheca [5,10,11,29–32], LET-413 acts upstream of DLG-1 in the epidermis.

As loss of HMR-1 and DLG-1 caused distinct phenotypes, we more closely examined the localization of these proteins in the seam cells, and their overlap with LET-413. All three pro-teins are present in a junctional band surrounding each seam cell, but HMR-1 is additionally present in the filopodia-like extensions (Fig 6C and 6K). This indicates that DLG-1 and LET-413 are located just basal to the protrusions, which is consistent with the more basal position-ing of the DAC relative to the CCC within the CeAJ [31,57]. We also investigated the func-tional relationship between HMR-1 and DLG-1 in the seam cells. Depletion of HMR-1 did not cause a loss of the DLG-1 junctional band (S5A and S5B Fig). We examined the effects of DLG-1 loss on HMR-1 at two timepoints: 4 h post-hatching, before the seam cell divisions, and 10 h post-hatching, when the seam cells in control animals have divided and reattached. As DLG-1 depletion is not complete, we also quantified DLG-1 levels in the same cells. Despite strong depletion of DLG-1, we still observed enrichment of HMR-1 at the subapical domain of the seam cells, at levels comparable to non-auxin-treated controls (S5C and S5D and S5E and S5F and S5G and S5H and S5I Fig). Thus, the localization of the DLG-1/AJM-1 and cadherin-catenin complexes in the seam cells are independent of each other, as previously shown in embryonic epithelia [5,10,11,30].

The findings above indicate that regulating the localization of DLG-1 is an essential func-tion of LET-413 that is required for seam cell outgrowth. Nevertheless, the phenotypes caused by DLG-1 depletion were somewhat less severe than those caused by LET-413 depletion. First, depletion of DLG-1 did not cause a growth arrest, though animals did appear to develop more slowly and became Dumpy. Additionally, the seam cells still formed small apical protrusions and/or small blebs (Fig 6G, arrows and S2 Video). The finding that depletion of LET-413 results in a more complete loss of protrusive activity than depletion of DLG-1 may be due to the incomplete depletion of DLG-1 but may also point to additional activities of LET-413 in promoting seam cell outgrowth. Taken together, our data show that DLG-1 is essential for directed seam cell outgrowth and indicate that LET-413 regulates seam cell outgrowth in large part by promoting the assembly or stability of DLG-1 at the CeAJ.

## Discussion

The cortical polarity protein Scrib plays conserved roles in promoting basolateral identity and junction assembly in epithelial cells, and in regulating cell proliferation and migration. Studies

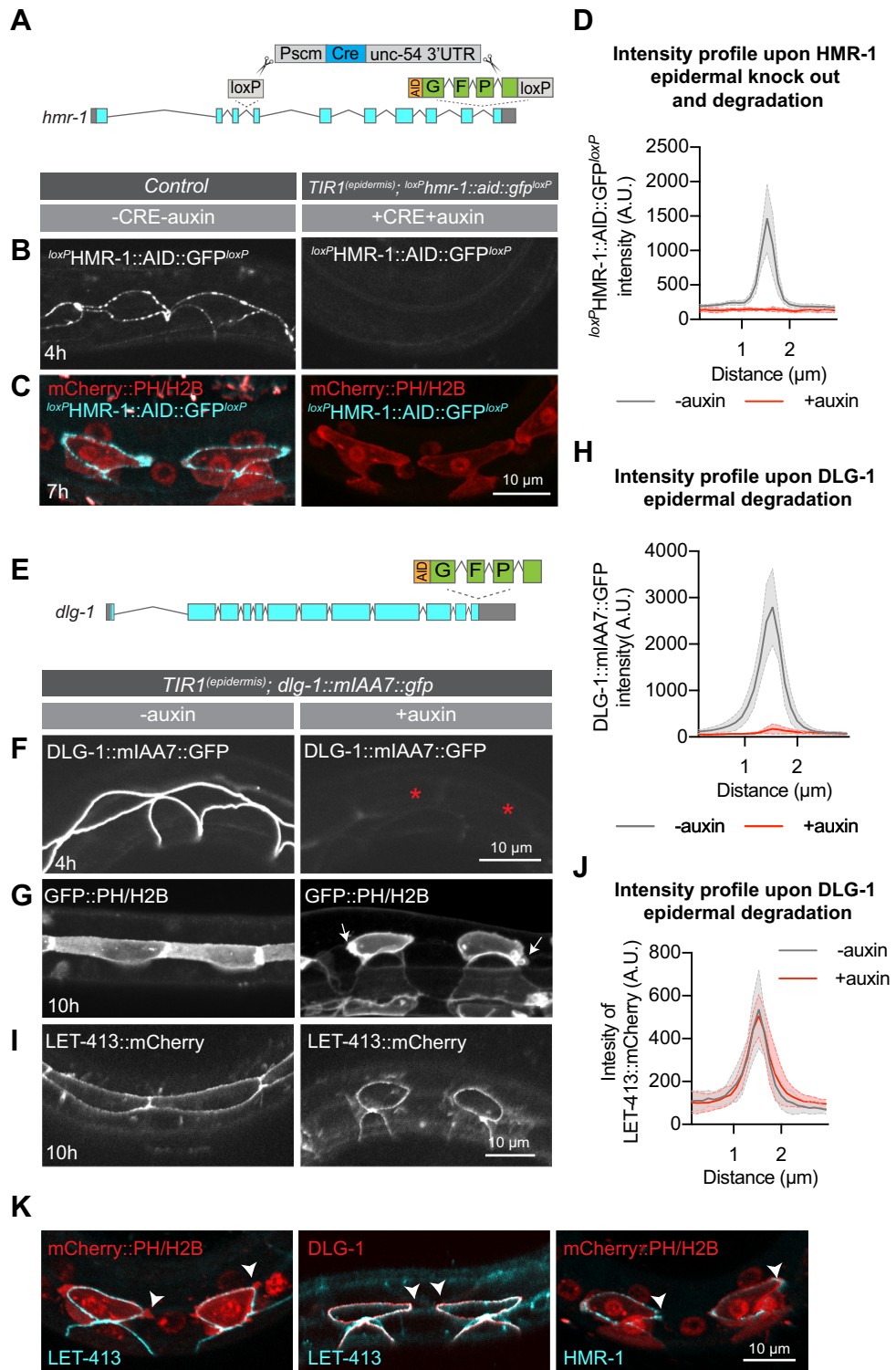

**Fig 6. LET-413 acts upstream of DLG-1 in the regulation of the seam cell outgrowth.** (**A**) Schematic representation of *hmr-1* depletion approach used (J. Cravo and S. van den Heuvel, manuscript in preparation). (**B, C**) Distribution of HMR-1 and the mCherry::PH and mCherry::H2B markers in the seam cells of control and HMR-1-depleted animals, 4 and 7 h after hatching. Images are maximum projections of the apical domain of the seam cells. Genotypes are *hmr-1::LoxP::AID::GFP::loxP; Pwrt2::mCherry::PH Pwrt-2::mCherry::H2B* for the control (strain SV2496) and *hmr-1::LoxP::AID::GFP::loxP; Pscm::CRE; Pwrt-2::TIR1::BFP; Pwrt2::GFP::PH Pwrt2::GFP::H2B* for the auxin-treated animals (strain SV2479). (**D**) Quantification of HMR-1 intensity across the hyp7–seam junction in animals depleted of HMR-1

as in B, C (+ auxin), and in control animals (- auxin). Graph shows mean GFP fluorescence intensity ± 95% CI. N = 7 animals for control and 3 animals for the HMR-1-depleted conditions. (**E**) Schematic representation of the insertion of GFP::AID-encoding sequences in the *dlg-1* locus. (**F, G**) Distribution of DLG-1 and GFP::PH and GFP::H2B markers, in control animals and DLG-1-depleted animals at indicated times post hatching. Genotypes are *Pwrt-2::TIR1::BFP; dlg-1::mIAA7::GFP* for panel F (strain BOX824), *Pwrt-2::TIR1::BFP; Pwrt2::GFP::PH Pwrt-2::GFP::H2B; dlg-1::mIAA7:: GFP* for panel G (strain BOX804). Images are maximum projections of the apical domain. Asterisks indicate the position of DLG-1-depleted seam cells. Arrows indicate small apical blebs, which were observed in one or more seam cells in 5/6 animals examined. (**H**) Quantification of DLG-1 intensity across the hyp7–seam junction in animals depleted of DLG-1 as in panels B, C. Graphs show mean apical GFP fluorescence intensity ± SD. N = 5 animals for -auxin, and 7 animals for +auxin. (**I**) Distribution of LET-413 in control animals and DLG-1-depleted animals at indicated times post hatching (strain BOX824). (**J**) Quantification of LET-413 intensity across the hyp7–seam junction in animals depleted of DLG-1 as in panels B, C. Graphs show mean apical GFP fluorescence intensity ± SD. N = 5 animals for -auxin and +auxin conditions. (**K**) Distribution of LET-413, DLG-1, and HMR-1 endogenously tagged with indicated fluorescent proteins in the extending seam cells. Arrowheads indicate the protrusions of the seam cells. Strains used, from left to right, are BOX531, BOX530, and SV2496.

of the single *C. elegans* Scrib homolog LET-413 identified essential roles for LET-413 in junction assembly and apical–basal polarization in the embryo and spermatheca, as well as a role in endocytic recycling in the intestine [5,10,11,29–33]. However, no essential roles in larval development or cell migration have been reported. Here, we used auxin-inducible protein degradation to bypass embryonic requirements and inactivate LET-413 in larval tissues. Using this approach, we find that expression of LET-413 in the larval epidermis is essential for growth and viability, and to promote directed outgrowth of the epithelial seam cells.

## LET-413 in growth and viability

The ubiquitous depletion of LET-413 from hatching onward caused a near complete lack of animal growth and resulted in larval lethality. Epidermal-specific depletion of LET-413 resulted in similar levels of larval lethality and a larval growth defect that was only slightly less severe. In contrast, we observed no growth defects or lethality when we depleted LET-413 in the intestine, consistent with previous observations of an intestine-specific CRISPR/Cas9 mutant of *let-413* [33]. Thus, the growth and viability defects appear to be largely due to an essential requirement for LET-413 in the epidermis, while LET-413 is not required in the intestine for larval development or viability. The more severe larval growth defects observed upon ubiquitous LET-413 depletion may reflect a minor contribution of additional tissues. Alternatively, it remains possible that TIR1 driven by the ubiquitous *eft-3* promoter results in more effective epidermal LET-413 depletion than TIR1 driven by the epidermal *wrt-2* promoter, even though we did not observe a difference in depletion by microscopy.

The underlying cause of the growth arrest following epidermal LET-413 degradation will require further investigation. It is not likely to be connected to the seam cell outgrowth defects, as the growth arrest is visible when the first seam cell divisions have not yet taken place. We recently found that epidermal depletion of the apical polarity regulators PAR-6 or PKC-3 similarly results in a larval growth arrest [37]. Thus, there may be a general requirement for apical–basal polarity regulators in the epidermis to support growth. Whether the growth defects are due to a loss of epidermal polarity or represent different functions of these polarity regulators remains to be determined.

## LET-413 in seam cell outgrowth

In LET-413-depleted animals, the posterior daughters of asymmetric seam cell divisions do not extend protrusions towards their neighboring cells, and consequently fail to reattach to each other. The process of directed seam cell extension has not been extensively studied. To our knowledge, the only genes specifically implicated in this process to date are *cdc-42* and

*nhr-25* [51,58]. CDC-42 is discussed further below. NHR-25 is a nuclear receptor family transcription factor whose inactivation causes a similar seam cell extension defect as LET-413 depletion [58]. Transcriptional targets through which NHR-25 controls seam cell outgrowth have not been identified, and *nhr-25* has numerous roles in *C. elegans* development in addition to seam cell outgrowth [59–62]. Nevertheless, ChIP-seq experiments have identified an NHR-25 binding site in the *let-413* promoter region [63], indicating a potential role for NHR-25 in regulating the transcription of *let-413*.

Because of the strong growth defects caused by LET-413 depletion, which are already apparent at the time of the first seam cell division, we considered whether the outgrowth defects are a secondary consequence of the growth arrest. However, we do not think that this is the case due to the highly specific nature of the seam cell phenotype. Division, fusion, and cell outgrowth all occur within a short time span, with outgrowth overlapping in time with the fusion process. Yet only outgrowth is affected by LET-413 depletion, and division and fusion take place as normal. Moreover, we did not observe a growth arrest in DLG-1-depleted animals, in which seam cells similarly fail to extend and reattach.

In a previous study using seam-specific RNAi, inactivation of *let-413* or the junction components *ajm-1* or *dlg-1* was postulated to cause a loss of seam cell fate and inappropriate fusion with the surrounding hypodermis, based on loss of an AJM-1::mCherry marker [64]. In our experiments with a PH::GFP membrane marker driven by the *wrt-2* promoter, we never observed fusion of posterior seam cells upon depletion of LET-413 or DLG-1. Nevertheless, depletion of LET-413 did not result in a complete loss of cell junctions nor in a complete loss of DLG-1 junction localization. It remains possible therefore that a stronger loss of junctional components could result in inappropriate fusion with the hypodermis. Alternatively, the observed difference could be due to differences in the timing of protein depletion by RNAi and auxin-inducible degradation.

LET-413 depletion also did not appear to affect the differentiation of the anterior daughter cells, as these fused with the hypodermis with normal kinetics. The only exception we observed followed the L2 seam cell divisions, with depletion of LET-413 starting in late L1 larvae. The division pattern in the L2 stage differs from the L1, L3, and L4 stages in that the asymmetric cell divisions are preceded by a symmetric division that increases the seam cell number. Thus, following the asymmetric, second division, four seam cells are generated, of which two will fuse with the hypodermis. At the time when fusion and reattachment of remaining seam cells were completed in control animals, the anterior seam cell daughters in LET-413 depleted animals had not fused. Possibly, LET-413 depletion affects aspects of seam differentiation that are particularly essential during the alternate L2 stage division pattern. Finally, the P cells in LET-413 depleted animals showed a reduced expression of fluorescent marker proteins driven by the *wrt-2* promoter. This may indicate a change in cell fate but could also be a secondary consequence of the failure in ventral retraction. Taken together, the loss of LET-413 may contribute to epidermal cell fate specification, but the primary defects we observe are in cell outgrowth and junctional integrity.

## Actin and Rho GTPase family members in seam cell outgrowth

The reported roles in some mammalian cell types for Scribble in controlling cell migration through the small GTPases Rac and Cdc42 [7,28] prompted us to consider a similar mode of action for LET-413. *C. elegans* Rho-family GTPases have already been shown to be involved in many migratory events, such as long-range migration of the Q neuroblasts and the gonadal distal tip cells [44,52,65], morphogenetic changes in embryonic epidermal cells during dorsal intercalation and ventral closure [49,54,66–69], and growth cone migration in axonal

pathfinding [52,53,70]. In addition, RNAi-mediated inactivation of *cdc-42* was shown to prevent seam cell outgrowth [51].

Our time-lapse and actin imaging data support an important role for the regulation of actin dynamics in seam cell outgrowth. Indeed, overexpression of constitutively active variants of CED-10 Rac or CDC-42 resulted in increased and undirected protrusive activity as well as seam extension failures, indicating that the spatial activity of these proteins needs to be carefully regulated for proper seam cell outgrowth and reattachment. However, we only achieved a partially penetrant and incomplete block in the formation of seam cell protrusions through mutation, RNAi, or expression of dominant negative variants of Rac family members and *cdc-42*. One possibility is that our inactivation experiments did not result in sufficient deregulation of Rac and Cdc42 signaling to disrupt seam cell extension. *C. elegans* expresses three Rac-family members, and in other migratory cell types redundancies have been shown between Rac family members themselves, as well as between Rac proteins and CDC-42 [52,54,70–72]. A further complicating factor is that the small GTPases play numerous essential roles in *C. elegans* development that may cause the most severely affected animals to arrest before reaching the stage of seam cell extension. Thus, determining whether regulation of actin dynamics by LET-413 plays a role in seam cell outgrowth remains an important future challenge.

## DLG-1 and cell junctions in seam cell outgrowth

The loss of LET-413 in the epidermis led to impaired cell junctions, evidenced by the highly fragmented appearance of the junction components DLG-1 Discs large and HMR-1 E-cadherin. Similar junctional defects were demonstrated in embryonic epidermal cells and the spermatheca upon *let-413* inactivation [5,10,11,29–32]. We did not observe junctional defects in the intestine, indicating that junction maintenance in this tissue does not require the continued expression of LET-413. In a previous study, intestine-specific CRIPSPR/Cas9-mediated knockout of *let-413* was reported to cause lateral displacement of HMP-1 α-catenin [33]. However, the *vha-6* promoter used to express Cas9 in this study is active during embryonic intestinal development, and the junctional defects may reflect a requirement in intestinal development. Our data show that LET-413 acts upstream of DLG-1 in junction maintenance in the seam epithelium, again in accordance with observations in the embryo and spermatheca [5,10,11,29–32]. In *Drosophila* different localization hierarchies between Scrib and Dlg have been described. In the adult *Drosophila* midgut epithelium, Scrib is required for the localization of Dlg [73], while in the ovarian follicle epithelium, Dlg acts upstream of Scrib [74,75]. In the embryonic epidermis, Scrib and Dlg are co-dependent for protein localization [14]. These differences may in part reflect the different subcellular locations of Dlg and Scrib in these tissues. In the midgut and embryonic epithelia, Dlg and Scrib localize to the septate junctions [14,73], while in the follicle epithelium, Dlg and Scrib have a broader basolateral localization pattern [74,75]. Thus, while LET-413 acts upstream of DLG-1 in all *C. elegans* tissues examined to date, it remains possible that future studies of additional tissues uncover a different relationship between these proteins.

One potential difference between our study and previous studies of *let-413* is the effect of LET-413 loss on apical–basal polarity. In embryonic epithelia, inactivation of *let-413* results in the basolateral invasion of apical proteins including PAR-3, PAR-6, and IFB-2 [11,29]. In the spermatheca, *let-413(RNAi)* also resulted in a lack of apical PAR-3. In contrast, we did not observe relocalization of the apical polarity protein PAR-6 or the basolateral polarity regulator LGL-1 upon LET-413 depletion. One explanation for this difference is that we investigate different tissues, and the epidermis may be less reliant on LET-413 for apical–basal polarization. We can also not rule out minor changes in localization not detectable with the markers and

microscopy approaches used. The most likely explanation, however, is that the difference is caused by the timing of inactivation. While previous studies inactivated *let-413* from the start of embryonic development or prior to development of the spermatheca, we depleted LET-413 only after epidermal tissues are established. While the seam cells do continue to divide, established tissues may be less reliant on LET-413 expression for maintaining apical–basal domain identity.

Interestingly, directed outgrowth of the seam cells appears to depend on the DLG-1/AJM-1 junctional complex, as depletion of DLG-1 resulted in a lack of seam cell outgrowth. What could the role of cell junctions in this process be? Cell–cell junctions are essential in transducing mechanical forces generated by actomyosin contractions into coordinated morphogenetic changes during cell–cell intercalation and collective cell migration [76–79]. The seam cells are somewhat similar to intercalating cells and cells undergoing collective migration in that they maintain their overall apical–basal polarity and cell junctions during outgrowth. However, force transduction is generally mediated by cadherin-based adherens junctions, while our experiments indicate that HMR-1 E-cadherin is dispensable for the directed outgrowth of the seam cells. A more likely possibility is that the DLG-1/AJM-1 junctional complex is involved in organizing the protrusive activity at the apical domain. Potential mechanisms for such spatial restriction include functioning as a hub for signaling components or promoting actin enrichment. For example, in the intestinal epithelium, DLG-1 has been shown to be required for the apical enrichment of actin [80]. Finally, we cannot exclude that DLG-1 has additional roles not related to cell junctions. For example, recent data in *Drosophila* point to a role for Dlg in transcriptional regulation by a nuclear pool of Dlg [81]. Regardless of the mechanism, our results demonstrate an essential role for LET-413 Scrib and DLG-1 Discs large in regulating the outgrowth of the epithelial seam cells.

## Materials and methods

### *C. elegans* strains and culture conditions

*C. elegans* strains were cultured under standard conditions [82]. Only hermaphrodites were used, and all experiments were performed with animals grown at 20°C on Nematode Growth Medium (NGM) agar plates. Table 1 contains a list of all the strains used.

### CRISPR/*Cas9* genome engineering

All gene editing made use of homology-directed repair of CRISPR/Cas9-induced DNA double-strand breaks. Final genomic sequences are available in S1 File. The sgRNA sequences and primers used to check integration can be found in Table 2. The *GFP::AID::let-413*, *hmr-1::GFP*, and *let-413::mCherry* edits were generated using plasmid-based expression of Cas9 and sgRNAs and plasmid repair templates containing 190–600 bp homology arms and a self-excising cassette (SEC) for selection of candidate integrants [83]. The *GFP::AID::let-413* and *hmr-1::GFP* repair templates were cloned using SapTrap assembly into vectors pDD379 and pMLS257 [84,85]. The *let-413::mCherry* repair template was cloned using Gibson assembly into vector backbone pJJR83 (Addgene #75028) [86,87]. All sgRNAs were expressed from plasmids under control of a U6 promoter. To generate *GFP::AID::let-413*, the sgRNAs were incorporated into assembly vector pDD379 using SapTrap assembly. For all other sgRNAs, antisense oligonucleotide pairs were annealed and ligated into BbsI-linearized pJJR50 (Addgene #75026) [88]. Injection mixes were prepared in MilliQ $H_2O$ and contained 50 ng/ml *Peft-3::cas9* (Addgene ID #46168) [89] 50–100 ng/ml U6::sgRNA, and 50–75 ng/ml of repair template. All mixes also contained 2.5 ng/ml of the co-injection pharyngeal marker *Pmyo-2::GFP* or *Pmyo-2::tdTomato* to aid in visual selection of transgenic strains. Young adult hermaphrodites were injected in

**Table 1. List of strains.**

| Strain | Genotype |
|---|---|
| N2 | Wild type |
| BOX043 | mibIs25 [cdc-42::GFP-2TEV-Avi 10ng + Pmyo-3::mCherry 10ng + lambda DNA 60ng] X |
| BOX245 | let-413(mib29[let-413::mCherry loxP]) V |
| BOX260 | dlg-1(mib35[dlg-1::AID::GFP loxP]) X |
| BOX449 | mibIs49[Pwrt-2::TIR1::mTagBFP2 lox511 tbb-2 3'UTR, IV:5014740–5014802 (cxTi10882 site)]) IV; let-413(mib81[GFP::loxP::AID::let-413]) V |
| BOX466 | let-413(mib81[GFP::loxP::AID::let-413]) V |
| BOX468 | let-413(mib81[GFP::loxP::AID::let-413]) V; mibIs48[Pelt-2::TIR1::mTagBFP2-lox511::tbb-2-3'UTR, IV:5014740–5014802 (cxTi10882 site)]) IV |
| BOX469 | ieSi57[Peft-3::TIR1::mRuby::unc-54 3'UTR + cb-unc-119(+)] II; unc-119(ed3) III; let-413(mib81[GFP::loxP::AID::let-413]) V |
| BOX527 | mibIs49[Pwrt-2::TIR1::mTagBFP2 lox511 tbb-2 3'UTR, IV:5014740–5014802 (cxTi10882 site)]) IV; let-413(mib29[let-413::mCherry LoxP]) V; dlg-1(mib35[dlg-1::AID::GFP-LoxP]) X |
| BOX530 | mib48[Pelt-2::TIR1::mTagBFP2-lox511::tbb-2-3'UTR]) IV; let-413(mib81[GFP::loxP::AID::let-413]) V; dlg-1(mib23[dlg-1::mCherry loxP]) X |
| BOX531 | let-413(mib81[GFP::loxP::AID::let-413]) V; huIs166[Pwrt-2::mCherry::PH + Pwrt-2::mCherry::H2B] X |
| BOX582 | mibIs49[Pwrt-2::TIR1::mTagBFP2 lox511 tbb-2 3'UTR, IV:5014740–5014802 (cxTi10882 site)]) IV; let-413(mib81[GFP::loxP::AID::let-413]) V; heIs63[Pwrt-2::GFP::PH + Pwrt-2::GFP::H2B + Plin-48::mCherry]V |
| BOX584 | hmr-1(he298[hmr-1::GFP loxP]) I; mibIs49[Pwrt-2::TIR1::mTagBFP2 lox511 tbb-2 3'UTR, IV:5014740–5014802 (cxTi10882 site)]) IV; let-413(mib81[GFP::loxP::AID::let-413]) V |
| BOX585 | mibIs49[Pwrt-2::TIR1::mTagBFP2 lox511 tbb-2 3'UTR, IV:5014740–5014802 (cxTi10882 site)]) IV; let-413(mib81[GFP::loxP::AID::let-413]) V; dlg-1(mib23[dlg-1::mCherry loxP]) X |
| BOX625 | mibIs49[Pwrt-2::TIR1::mTagBFP2 lox511 tbb-2 3'UTR, IV:5014740–5014802 (cxTi10882 site)]) IV; let-413(mib81[GFP::loxP::AID::let-413]) V; heIs63[Pwrt-2::GFP::PH + Pwrt-2::GFP::H2B + Plin-48::mCherry] V; mcIs40 [Plin-26::ABDvab-10::mCherry + Pmyo-2::GFP] |
| BOX692 | par-6(mib24[par-6::eGFP LoxP] I; mibIs49[Pwrt-2::TIR1::mTagBFP2 lox511 tbb-2 3'UTR, IV:5014740–5014802 (cxTi10882 site)]) IV; let-413(mib81[GFP::loxP::AID::let-413]) V; mibEx251[Pwrt-2::mCherry::H2B + Pwrt-2::mCherry::PH + Plin-48::tdTomato] |
| BOX693 | par-6(mib24[par-6::eGFP LoxP]) I; let-413(mib29[let-413::mCherry LoxP]) V; mibEx251[Pwrt-2::mCherry::H2B 60ng + Pwrt-2::mCherry::PH 60 ng+ Plin-48::tdTomato 15ng + lambda DNA 50ng] |
| BOX694 | mibIs49[Pwrt-2::TIR1::mTagBFP2 lox511 tbb-2 3'UTR, IV:5014740–5014802 (cxTi10882 site)]) IV; let-413(mib81[GFP::loxP::AID::let-413]) V; mibIs23[lgl-1::GFP::2xTEV::Avi 10 ng + Pmyo-3::mCherry 10 ng + lambda DNA 60 ng] V |
| BOX697 | heIs63[Pwrt-2::GFP::PH + Pwrt-2::GFP::H2B + Plin-48::mCherry] V; mibEx264[Pwrt-2::mcherry::T2A::ced-10(T17N) 30ng + PMyo-2::tdtomato 3ng + lambda DNA 47ng] |
| BOX699 | ced-10(n1993)/tmC25[unc-5(tmIs1241)] + Pmyo-2::Venus]; heIs63[Pwrt-2::GFP::PH + Pwrt-2::GFP::H2B + Plin-48::mCherry] V; mig-2(mu28)X |
| BOX700 | heIs63[Pwrt-2::GFP::PH + Pwrt-2::GFP::H2B + Plin-48::mCherry] V; mibEx253[wrt-2::mCherry::T2A::ced-10(Q61L):: tbb-2UTR 30ng + Myo2::tdtomato 3ng + lambda DNA 47ng] |
| BOX739 | heIs63[Pwrt-2::GFP::PH + Pwrt-2::GFP::H2B + Plin-48::mCherry] V; mibEx265[Pwrt2::cdc-42(Q61L):: F2A::BFP::NLS::tbb-2UTR 40ng + PMyo-2::tdtomato 3ng + lambda DNA 37ng] |
| BOX747 | heIs63[Pwrt-2::GFP::PH + Pwrt-2::GFP::H2B + Plin-48::mCherry] V; mibEx266[Pwrt2::cdc-42(T17N):: F2A::BFP::NLS::tbb-2UTR 40ng + PMyo-2::tdtomato 3ng + lambda DNA 37ng] |
| BOX804 | mibIs49[Pwrt-2::TIR-1::tagBFP2-Lox511::tbb-2-3'UTR, IV:5014740–5014802 (cxTi10816 site)]) IV; heIs218[Pwrt-2::mCherry::PH, Pwrt-2::mCherry::H2B, Plin-48::GFP]; dlg-1(mib163[dlg-1::IAA7[37–104](co)::GFP(co)]) X |
| BOX824 | mibIs49[Pwrt-2::TIR-1::tagBFP2-Lox511::tbb-2-3'UTR, IV:5014740–5014802 (cxTi10816 site)]) IV; let-413(mib29[let-413::mCherry-LoxP]) V; dlg-1(mib163[dlg-1::IAA7[37–104](co)::GFP(co)]) X |
| BOX825 | hmr-1(he298[hmr-1::egfp::loxP]) I; mibIs49[Pwrt-2::TIR-1::tagBFP2-Lox511::tbb-2-3'UTR, IV:5014740–5014802 (cxTi10816 site)]) IV; dlg-1(mib167[dlg-1::IAA7[37–104](co)::mScarlet(co)]) X |
| BOX826 | mibIs49[Pwrt-2::TIR-1::tagBFP2-Lox511::tbb-2-3'UTR, IV:5014740–5014802 (cxTi10816 site)]) IV; heIs63 [Pwrt-2::GFP::PH + Pwrt-2::GFP::H2B + Plin-48::mCherry] V; dlg-1(mib167[dlg-1::IAA7[37–104](co)::mScarlet(co)]) X |

(*Continued*)

**Table 1.** (Continued)

| Strain | Genotype |
|---|---|
| BOX832 | hmr-1(he377[hmr-1::LoxP::aid::gfp::LoxP] I; heSI175[Pscm::CRE] X; mib69[Pwrt-2::TIR-1::BFP] IV; dlg-1(mib159[dlg-1::mCherry(co)]) X |
| CA1200 | unc-119(ed3); ieSi57 [Peft-3::TIR1::mRuby unc-54 3'UTR + cb-unc-119(+)] II |
| FT1459 | xnIs506 [Pcdc42::GST::GFP::wsp-1(GBD) + unc-119(+)] |
| KN2598 | huIs166[Pwrt-2::mCherry::PH + Pwrt-2::mCherry::H2B] X |
| SV1009 | heIs63[Pwrt-2::GFP::PH + Pwrt-2::GFP::H2B + Plin-48::mCherry] V |
| SV1550 | heIs1175 [Pscm::CRE] X |
| SV1955 | hmr-1(he298[hmr-1::GFP loxP]) I |
| SV1984 | heIs218[Pwrt-2::mCherry::PH + Pwrt-2::mCherry::H2B + Plin-48::GFP] |
| SV2239 | hmr-1(he377[hmr-1::loxP::AID::loxP]) I |
| SV2475 | mibIs49[Pwrt-2::TIR1::mTagBFP2 lox511 tbb-2 3'UTR, IV:5014740–5014802 (cxTi10882 site)]) IV; dlg-1(he380[dlg-1::loxP::AID::GFP::loxP]) X; huIS166[Pwrt-2::mCherry::H2B + Pwrt-2::mCherry::PH] X |
| SV2479 | hmr-1(he377[hmr-1::loxP::AID::GFP::loxP]) I; heIs175[Pscm::CRE] X; mibIs49[Pwrt-2::TIR1::mTagBFP2 lox511 tbb-2 3'UTR, IV:5014740–5014802 (cxTi10882 site)]) IV; heEx616[Pwrt-2::mCherry::H2B + Pwrt-2::mCherry::PH + Plin-48::tdTomato + Prps-0::HygR + lambdaDNA] |
| SV2496 | hmr-1(he298[hmr-1::GFP loxP]) I; heIs218[Pwrt-2::mCherry::PH + Pwrt-2::mCherry::H2B + Plin-48::GFP] IV |

**Table 2. Reagents to generate CRISPR/Cas9 lines.**

| | |
|---|---|
| **GFP::AID::LET-413** | |
| sgRNA sequence | gcagaagaaagccggcattgTGG |
| Integration check primer left | 5'-CGGTGTCACCTACGCCTAAT |
| Integration check primer right | 5'-GCTTCGAGAACTCGCAGATTC |
| **DLG-1::mIAA7::GFP, DLG-1::mIAA7::mScarlet, and DLG-1::mCherry** | |
| sgRNA sequence | gccacgtcattagatgaaatTGG |
| Integration check primer left | 5'-CAGTAGCTGCGTTCCACGTA |
| Integration check primer right | 5'-TCATCAGTAGAGAGTCGGGAATATG |
| mIAA7::GFP Sp9 oligo F | GTGAATCGCAGACGCCAATTTGGGTGCCACGTCATggcggtgggggaGGATTCTCCGAGACCGTCG |
| mIAA7::GFP Sp9 oligo R | aaacgattatttgtctaaaaaatatccaatttcatCTACTTGTAGAGCTCGTCCATTCCG |
| mIAA7::mScarlet Sp9 oligo F | GTGAATCGCAGACGCCAATTTGGGTGCCACGTCATggcggtgggggaGGATTCTCCGAGACCGTCG |
| mIAA7::mScarlet Sp9 oligo R | aacgattatttgtctaaaaaatatccaatttcatCTACTTGTAGAGCTCGTCCATTCCTC |
| mCherry Sp9 oligo F | GTGAATCGCAGACGCCAATTTGGGTGCCACGTCATggcggtgggggaATGTCCAAGGGAGAGGAGG |
| mCherry Sp9 oligo R | cgattatttgtctaaaaaatatccaatttcatCTACTTGTAGAGCTCGTCCATTCCTC |
| **LET-413::mCherry** | |
| sgRNA sequence | agtaggccatgtgagtattgAGG |
| Integration check primer left | 5'-CTTGCCGCAGGCACTCAAAA |
| Integration check primer right | 5'-GTTGTGTGAGCTCATGAGAGTTGGG |
| **HMR-1::GFP and HMR-1::AID::GFP (with first LoxP site)** | |
| sgRNA sequence | cgaaagtgcccaataaacgaCGG |
| Integration check primer left | cgtggcctacattacatgca |
| Integration check primer right | tgttacgtctcgcatgccta |
| **LoxP-HMR-1::AID::GFP-LoxP (second LoxP site addition)** | |
| sgRNA sequence | gtgtcgaaatgcaccataatCGG |
| ssODN | gttttgaagcttgtgtcgaaatgcaccataATAACTTCGTATAGCATACATTATACGAAGTTATatcggaattctggttgagtttgtgaaaat |
| Integration check primer left | ctcctccacctctgtcgttttc |
| Integration check primer right | ttcacaaccccccaaatccat |

the germline using an inverted micro-injection setup (Eppendorf FemtoJet 4x mounted on a Zeiss Axio Observer A.1 equipped with an Eppendorf Transferman 4r). Candidate edited progeny were selected on plates containing 250 ng/ml of hygromycin and correct genome editing was confirmed by Sanger sequencing (Macrogen Europe) of PCR amplicons encompassing the edited genomic region. From correctly edited strains, the hygromycin selection cassette was excised by a heat shock of L1 larvae at 34˚C for 1 h in a water bath.

The *LoxP::hmr-1::AID::GFP::LoxP* allele was generated using plasmid-based expression of Cas9 and sgRNAs in two stages. In the first stage, *AID::GFP::LoxP* was inserted using a plasmid repair template cloned using SapTrap into vector pMLS257, using the procedure described above. In the second stage, the second *LoxP* site was inserted using a single strand DNA repair oligo (Table 2) and PCR based selection of correct inserts.

The *dlg-1::mIAA7::GFP*, *dlg-1::mIAA7::mScarlet*, and *dlg-1::mCherry* alleles were generated using the Alt-R CRISPR/Cas9 system (IDT). Repair templates were synthesized with 5' Sp9-modified oligos (IDT) and Q5 polymerase (NEB) from plasmids pJJS001 (mIAA7 GFP), pRS188 (mIIA7 mScarlet) and pRS066 (mCherry), and purified using the MinElute PCR Purification kit (Qiagen). See Table 2 for oligo's used and S1 File for plasmid templates. Injection mixes were prepared with melted dsDNA repair templates as described [90]. Positive F1 from injected P0 animals were selected based on fluorescence. Correct genome editing was confirmed by Sanger sequencing (Macrogen Europe) of PCR amplicons encompassing the edited genomic region.

## CED-10 and CDC-42 Q61L and T17N expression plasmids

Plasmids for expression of constitutively active and dominant negative variants of CED-10 and CDC-42 were generated using Gibson assembly. Final plasmid sequences are available in S1 File. CED-10b and CDC-42 coding sequences carrying the appropriate base pair changes were ordered as gBlocks Gene Fragments (Integrated DNA Technologies). *cdc-42* was codon optimized [91], while *ced-10* was not optimized due to inability to synthesize the gBlock for an optimized variant. Plasmids contain 2A self-cleaving peptide sequences [92] between GTPase and fluorophore, to enable visual identification of expressing cells while minimizing the risk of altering the activity of the GTPase. Plasmids contain homology arms for insertion at the cxTi10816 Mos transposon site on chromosome IV but were only used as extrachromosomal arrays. For the generation of the lines, 20–30 ng/ul of the plasmids were injected with 50 ng/ul of lambda DNA and co-injection marker *Pmyo-2::tdTomato* (2,5 ng/ul).

## Animal synchronization and auxin treatment

NGM + auxin plates were prepared using the natural indole-3-acetic acid (IAA) from Alfa Aesar (#A10556). The stock powder was stored at 4˚C and was diluted in NGM agar cooled to 50˚C to a final concentration of 3 mM. The plates were left to dry at room temperature for 1–2 days covered with aluminum foil before seeding with OP50 bacteria. The plates were then kept at room temperature for another 1–2 days before storing at 4˚C in the dark for a maximum of 2 weeks. For auxin treatment in liquid M9, the water-soluble synthetic analog of IAA, 1-naphthaleneacetic acid (NAA), from Sigma Aldrich (#317918) was used. A 250 mM stock solution was prepared in M9 buffer and was further diluted in M9 buffer to 3 mM final concertation.

To synchronize animals, plates with eggs were washed with M9 buffer (0.22 M KH2PO4, 0.42 M Na2HPO4, 0.85 M NaCl, 0.001 M MgSO4) to remove larvae and adults but leave the eggs behind. After 1 h, plates were washed again to collect larvae hatched within that time span. Synchronized larvae were then transferred onto NMG-OP50 plates with auxin. For the

depletion of DLG-1 in Fig 5, gravid adult animals were bleached, and eggs were hatched in M9. Newly hatched larvae were kept in M9 with auxin but without food for 24 h before transferring onto NMG-OP50 plates with auxin. For non-auxin treated controls, NGM plates and M9 buffer lacking auxin were used.

## Feeding RNAi

For feeding RNAi experiments, bacteria were pre-cultured in 2 ml Lysogeny Broth (LB) supplemented with 100 μg/ml ampicillin (Amp) and 2.5 μg/ml tetracycline (Tet) at 37˚C in an incubator rotating at 200 rpm for 6–8 h, and then transferred to new tubes with a total volume of 10 ml LB for overnight culturing. To induce production of dsRNA, cultures were incubated for 90 min in the presence of 1 mM Isopropyl β-D-1-thiogalactopyranoside (IPTG). Bacterial cultures were pelleted by centrifugation at 4000 g for 15 min and resuspended in LB with 100 μg/ml ampicillin (Amp) and 2.5 μg/ml tetracycline (Tet) at 5x the original concentration. NGM agar plates supplemented with 100 μg/ml Amp and 1 mM IPTG were seeded with 250 μl of bacterial suspension, and kept at room temperature (RT) for 48 h in the dark. Six to eight L4 hermaphrodites per strain were transferred to individual NGM-RNAi plates against target genes and phenotypes were analyzed in the F1 generation.

## Larval lethality

To determine larval lethality, synchronized L1 animals were placed on NGM plates seeded with *E. coli* OP50, and either containing or lacking auxin. After 24 h animals were classified as dead or alive based on movement and response to physical touch.

## Microscopy and image processing

Imaging of *C. elegans* on agar plates for growth analysis was done using a Zeiss Axio Zoom. V16 equipped with a PlanNeoFluar Z 1x/0.25 objective and an Axiocam 506 color camera, driven by Zen Pro software. All other imaging of *C. elegans* was done by mounting embryos or larvae on a 5% agarose pad in a 10 mM Tetramisole solution in M9 buffer to induce paralysis. Nomarski DIC imaging was performed with an upright Zeiss AxioImager Z2 microscope using a 63 x 1.4 NA objective and a Zeiss AxioCam 503 monochrome camera, driven by Zeiss Zen software. Spinning disk confocal imaging was performed using a Nikon Ti-U microscope driven by MetaMorph Microscopy Automation and Image Analysis Software (Molecular Devices) and equipped with a Yokogawa CSU-X1-M1 confocal head and an Andor iXon DU-885 camera, using 60x or 100x 1.4 NA objectives. All stacks along the z-axis were obtained at 0,25 μm intervals. Maximum intensity Z projections were done in ImageJ (Fiji) software [93,94]. For quantifications, the same laser power and exposure times were used within experiments. Time-lapse imaging of *C. elegans* seam cell membrane dynamics was performed on a Zeiss Lattice Lightsheet 7 pre-serial system equipped with 13.3x/0.44 excitation and 44.83x/1 observation lenses, 15x550, 15x650, 30x700, 300x1000, 100x1400 and 100x1800 sinc3 light sheets, 488/561/640 nm excitation lasers, BP 420-480/BP 495-550/LP 650, LP 655, BP 570-650/LP 750, BP 495-575/LP 750, LBF 405/488/561/642 nm emission filters, controlled by ZEN 3.4 (blue edition, Zeiss) software. Precision cover glasses thickness no. 1.5H (Marienfeld-superior) and 5% agarose in MQ H2O pads on glass slides (incubated for 5 minutes in M9 buffer) were used to mount synchronized L1 larvae in M9 buffer + 10 mM tetramisole. Images were acquired at a 20 s intervals with 0,2 um spacing (685 slices). Resulting data sets were deskewed and deconvolved using ZEN 3.4 (blue edition, Zeiss) software, and finally rendered using Vision4D 3.5 (arivis AG). Image scales were calibrated for each microscope using a micrometer slide. For display in figures, level adjustments, false coloring, and image overlays were done

in Adobe Photoshop. Image rotation, cropping, and panel assembly were done in Adobe Illustrator. All edits were done non-destructively using adjustment layers and clipping masks, and images were kept in their original capture bit depth until final export from Illustrator for publication.

## Quantitative image analysis

All quantifications were done using ImageJ.

***C. elegans* growth curves.** Synchronized L1 animals were placed on NGM plates seeded with *E. coli* OP50 and either lacking or containing 3 mM auxin, and images were taken after 3, 7, 24, and 30 h. Animal lengths were measured by drawing a spline along the center line of the animal.

**Fluorescence intensity measurements.** For all fluorescence intensity measurements, mean background fluorescence levels were subtracted from measured values. Mean background intensity was determined in a circular region of ~50 px diameter in areas within the field-of-view that did not contain any animals.

Distribution plots of fluorescence intensity of GFP::AID-tagged LET-413 in the larval intestine and epidermis, as well as AID::GFP-tagged DLG-1, GFP-tagged LGL-1, and mCherry-tagged LET-413 in the epidermis were obtained by averaging the peak values of intensity profiles from 2–4 10 px-wide line-scans perpendicular to the membrane per animal. In the auxin-treated samples, where LET-413 is no longer enriched at the membrane, DLG-1::mCherry or the PH marker were used to determine the position to quantify. The intensity profiles of different animals were aligned at their peak values and trimmed manually to exclude values outside the cells/compartments of interest. For the intensity of GFP-tagged PAR-6, peak values of intensity profiles from multiple 10 px-wide line-scans in the apical side of the cytoplasm of the seam cells were obtained, averaged, and corrected for the background of the same animal. In all cases the graphs indicate the mean intensities of the values.

## Statistical analysis

All statistical analyses were performed using GraphPad Prism 8. For population comparisons, a D'Agostino & Pearson test of normality was first performed to determine if the data was sampled from a Gaussian distribution. For data drawn from a Gaussian distribution, comparisons between two populations were done using an unpaired t test, with Welch's correction if the SDs of the populations differ significantly, and comparisons between >2 populations were done using a one-way ANOVA, or a Welch's ANOVA if the SDs of the populations differ significantly. For data not drawn from a Gaussian distribution, a non-parametric test was used (Mann-Whitney for 2 populations and Kruskal-Wallis for >2 populations). ANOVA and non-parametric tests were followed up with multiple comparison tests of significance (Dunnett's, Tukey's, Dunnett's T3 or Dunn's). Tests of significance used and sample sizes are indicated in the figure legends. No statistical method was used to pre-determine sample sizes. No samples or animals were excluded from analysis. The experiments were not randomized, and the investigators were not blinded to allocation during experiments and outcome assessment. Numerical data for graphs and summary statistics are available in spreadsheet form in S2 File.

## Supporting information

**S1 Fig. Larval expression of LET-413.** Expression of GFP::AID::LET-413 in the pharynx (left), reproductive system (middle), and excretory canal (right) of L4 stage *GFP::AID::let-413* animals (strain BOX466). Sp: spermatheca, Ut: Uterus, Vul: Vulva. Related to Fig 1.
(PDF)

**S2 Fig. LET-413 is required for seam cell outgrowth throughout development.** Time series of L2 and L3 seam cells divisions and subsequent extension in LET-413-depleted (+auxin) or control animals (-auxin) (strain BOX582). Seam-specific GFP::H2B and GFP::PH mark DNA and cell membrane, respectively. Times indicate hours post hatching. For the L2 division pattern (blue), auxin was added from 8h after hatching, and for the L3 division pattern (red) from 19h after hatching. Related to Fig 3.
(PDF)

**S3 Fig. Degradation of LET-413 in the epidermis does not affect Q cell migration. (A)** Q cell descendant QRa and QRb during anterior migration (4–5 h post hatching), marked with epidermal-specific mCherry::H2B and mCherry::PH (strain BOX531). No expression of GFP::AID::LET-413 is detected. **(B)** Migration and division of Q cell descendants in LET-413-depleted (+auxin) or control animals (-auxin) at 2–3 h and 4–5 h post hatching (strain BOX582). Related to Fig 3.
(PDF)

**S4 Fig. LET-413 depletion disrupts the localization of DLG-1.** Distribution of DLG-1::mCherry in the epidermis of *Pwrt-2::TIR1::BFP; GFP::AID::let-413; dlg-1::mCherry* animals without (-auxin) and in the presence of auxin (+auxin) at 5 h and 7 h post hatching (strain BOX527). Boxed region in top overview panels is shown enlarged below. Related to Fig 5. Note that 7 h timepoint is also shown in Fig 5E, and replicated here for ease of comparison between time points.
(PDF)

**S5 Fig. DLG-1 and HRM-1 localize independently in the epidermis.** (**A**) Distribution of DLG-1::mCherry upon depletion and degradation of HMR-1 at 10 h post hatching. Left panels are controls not expressing CRE and not treated with auxin, right panels show animals expressing CRE and treated with auxin. Genotypes are *hmr-1::LoxP::AID::GFP::loxP; Pwrt2::GFP::PH Pwrt-2::GFP::H2B* for the control (strain BOX832) and *hmr-1::LoxP::AID::GFP::loxP; Pscm::CRE; Pwrt-2::TIR1::BFP; Pwrt2::GFP::PH Pwrt-2::GFP::H2B* for the CRE and auxin-treated animals (strain BOX832). (**B**) Quantification of HMR-1 intensity across the hyp7–seam junction in animals depleted of HMR-1 as in A (+ auxin), and in control animals (-auxin). Graph shows mean mCherry fluorescence intensity ± 95% CI. N = 5 animals for control and 5 animals for the HMR-1-depleted conditions. (**C, F**) Distribution of HMR-1::GFP upon depletion of DLG-1 at 4 h (C) and 10 h (F) post hatching. Left panels are controls not treated with auxin, and right panels are animals in which DLG-1 was depleted by auxin treatment. Genotype is *hmr-1::GFP; Pwrt-2::TIR1::BFP; dlg-1::mIAA7::mCherry* (strain BOX825). (**D, E, H, I**) Quantification of DLG-1::mIAA7::mScarlet or HMR-1::GFP intensity across the hyp7–seam junction in animals depleted of DLG-1 as in C, F (+ auxin), and in control animals (- auxin). Graph shows mean mCherry fluorescence intensity ± 95% CI. N = 6 animals for control and 5 animals for the DLG-1-depleted conditions. Related to Fig 6.
(PDF)

**S1 Video. Time-lapse imaging of seam cell extension and fusion.** GFP::PH$^{PLC1\delta}$ and GFP::H2B mark the cell membrane and DNA (strain SV1009). Images were taken at 20 sec intervals at 7 h post hatching. Related to Fig 4.
(MP4)

**S2 Video. Time-lapse imaging of seam cells upon depletion of DLG-1.** GFP::PH$^{PLC1\delta}$ and GFP::H2B mark the cell membrane and DNA (strain BOX826). Images were taken starting at

7 h post hatching at 10 min. intervals. Maximum intensity projections are shown. In this video, left side is anterior and top is ventral. Related to Fig 6.
(MP4)

**S1 File. DNA files.** This.zip file contains DNA files of the genomic regions of all CRISPR engineered loci, of plasmids used as PCR templates for the generation of repair templates, and of dominant active and dominant negative constructs of CED-10 and CDC-42. Files are present in two formats: a binary SnapGene file, which can be opened with the SnapGene viewer, and a text-based Genbank file, which is compatible with all text editors but contains less extensive feature formatting.
(ZIP)

**S2 File. Numerical data for graphs and summary statistics.** Numerical data for graphs and summary statistics in Microsoft Excel format.
(XLSX)

# Acknowledgments

We thank S. Ruijtenberg for strain SV1550, M. Soto for the *gex-2* RNAi clone, and members of the S. van den Heuvel, S. Ruijtenberg, and M. Boxem groups for helpful discussions. We also thank Wormbase (Harris et al., 2020) and the Biology Imaging Center, Faculty of Sciences, Department of Biology, Utrecht University. Some strains were provided by the Caenorhabditis Genetics Center, which is funded by NIH Office of Research Infrastructure Programs (P40 OD010440).

# Author Contributions

**Conceptualization:** Amalia Riga, Janine Cravo, Sander van den Heuvel, Mike Boxem.

**Formal analysis:** Amalia Riga.

**Funding acquisition:** Sander van den Heuvel, Mike Boxem.

**Investigation:** Amalia Riga, Janine Cravo, Ruben Schmidt, Helena R. Pires, Victoria G. Castiglioni.

**Methodology:** Amalia Riga, Janine Cravo, Helena R. Pires.

**Project administration:** Mike Boxem.

**Supervision:** Sander van den Heuvel, Mike Boxem.

**Visualization:** Amalia Riga, Mike Boxem.

**Writing – original draft:** Amalia Riga, Mike Boxem.

**Writing – review & editing:** Janine Cravo, Ruben Schmidt, Helena R. Pires, Victoria G. Castiglioni, Sander van den Heuvel.

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
