## [Decision Letter · Decision Letter 0]

17 May 2021

Dear Dr Boxem,

Thank you very much for submitting your Research Article entitled 'Caenorhabditis elegans LET-413 Scribble is essential in the epidermis for growth, viability, and directional outgrowth of epithelial seam cells' to PLOS Genetics.

The manuscript was fully evaluated at the editorial level and by independent peer reviewers. The reviewers appreciated the attention to an important problem, but raised some substantial concerns about the current manuscript. In particular, several reviewers were concerned about the interpretation of experiments utilizing constitutively active Rho GTPases and recommended additional experiments to clarify the link between these proteins and LET-413. Based on the reviews, we will not be able to accept this version of the manuscript, but we would be willing to review a much-revised version. We cannot, of course, promise publication at that time.

If you decide to revise the manuscript for further consideration at PLOS Genetics, please aim to resubmit within the next 60 days, unless it will take extra time to address the concerns of the reviewers, in which case we would appreciate an expected resubmission date by email to plosgenetics@plos.org.

[LINK]

We are sorry that we cannot be more positive about your manuscript at this stage. Please do not hesitate to contact us if you have any concerns or questions.

Yours sincerely,

Jeremy Nance

Associate Editor

PLOS Genetics

Gregory P. Copenhaver

Editor-in-Chief

PLOS Genetics

Reviewer's Responses to Questions

**Comments to the Authors:**

Reviewer #1: The conserved protein Scribble is a scaffolding molecule that contributes to cell polarity, cell migration and tissue structure across animals. The C. elegans Scribble homolog let-413 is essential for embryonic morphogenesis. Since let-413 mutants die during embryogenesis, post-embryonic roles of let-413 have been unknown. Here, Boxem and colleagues describe phenotypes induced by depleting LET-413 post-embryonically via the auxin-inducible degron system. Interestingly, no clear defects were observed upon depletion of LET-413 in intestinal cells; however, LET-413 depletion produced defects in epidermal seam cells, which ultimately led to larval arrest and lethality.

This is a straightforward and interesting study that describes novel (to my knowledge) roles of LET-413 in the larval epidermis. The manuscript is well written and the data are, for the most part, clearly presented. I have some comments for the authors to consider, but no serious concerns that would preclude publication.

• I found the images and quantification in the first half of Figure 4 somewhat unconvincing.

o In Figure 4A, are we seeing a lateral or apical focal plane?

o In Figure 4B, I’m not sure a line scan is the best form of quantification. There is clearly still cortical PAR-6, but without knowing what focal plane we’re in, I don’t know how to interpret this.

o I like that Figure 4C shows both apical and lateral focal planes, but it’s hard to see what is going on because the signal:background is poor. While I generally applaud the use of endogenous tags for everything, this might be a rare case where a seam cell-specific GFP::PAR-6 transgene would give clearer results.

o For Figures 4D-E, were both apical and lateral planes acquired? If not, how do we know that LGL-1::GFP is not mislocalized apically?

o My suggestion would be to acquire closely-spaced Z stacks of these animals, and then show reconstructed YZ images (corresponding to a transverse section across the body of the animal) to clearly visualize both the apical and lateral membranes. Quantification could then directly compare apical vs. lateral fluorescence intensity.

• It’s difficult to see what’s going on in Figure 3E. Why is the P cell brightly labeled by GFP::PH in the -auxin but not the +auxin condition? If this reflects a trivial issue like choice of focal plane, I suggest the authors acquire new images that show the phenotype clearly.

• The authors could do a better job making the experimental conditions clear from their figure legends. For every figure panel, the figure panel should include 1) What strain is being imaged – ideally including the strain name so that the reader can cross-reference to the strain table; 2) What larval stage is being imaged; and 3) How many replicates the image represents. This information is present in some figure legends, but it’s not consistent throughout the paper.

Reviewer #2: Riga et al., show that inducible degradation can be used to overcome the need of Scrib (LET-413) during C.elegans embryogenesis to study tissue-specific functions during late larval development. This led to the identification of a new role for LET-413 in the polarized outgrowth of epidermal seam cells. The phenotype is very well characterized by live imaging to follow seam cell division pattern and the subsequent steps until showing the specific inability of seam cells to form protrusions. This finding is very interesting by providing the first evidence that Scrib plays a role in protrusive cell shape change in invertebrates, which can bring new parallels with the roles in cell migration described in vertebrates. The paper is generally well-written and given the broad relevance of the Scrib protein, it could gather general interest by cell and developmental biologists.

The manuscript also shows that another component of the Scrib module, Dlg, has a similar role in the regulation of protrusive activity and proposes that not only Dlg but also CDC42 and the CED-10 RAC act downstream of LET-413. The conclusions taken in this second part of the manuscript are not completely supported by the data. For instance, the manuscript provides evidences against a fundamental role of CDC42/RAC by showing that their inactivation does not affect the protrusive activity of seam cells. Moreover, compelling experiments linking the role of GTPases to LET-413 function are absent. Addressing this and some other specific points described below should be considered before publication.

MAJOR ISSUES

1) line 256 “ We also did not detect apical invasion or reduced basolateral levels of LGL-1 (Fig. 4D, E).” Legend states that this figure is a maximum projection of apical and junctional domain and so evidence of the maintenance of basolateral levels of LGL-1 should be provided by imaging planes similar to the ones in Fig. 4C for PAR6. Moreover, there is an increase in the peak of Lgl intensity after auxin treatment in the graph of Fig. 4E, which is intriguing and should be analyzed statistically to clarify if there is apical invasion.

2) The manuscript shows clearly that LET-413 function is both required for the organization of cadherin-based junctions and to maintain the normal localization pattern of Dlg at the junctional level (Fig. 4F-4G). This result helps defining the functional hierarchy of the Scrib module components. However, the finding that Dlg mislocalization only becomes apparent few hours after junction disruption brings in the alternative hypothesis that defects in junctional organization are the underlying cause of Dlg mislocalization. The authors should tackle experimentally this alternative or at least discuss it.

3) The manuscript suggests that DLG-1 acts downstream of LET-413 while showing a weaker phenotype of DLG-1 induced degradation. The evidence for the less dramatic effect is robust, but the phenotype should be further analyzed to clarify the relation with the phenotypes presented by LET-413 depletion. In particular, it would be worth to analyse HMR-1 localization to evaluate if it was equally affected by DLG-1 depletion. Considering if different effects in cadherin-based junctions could be related to the additional features of the LET-413 phenotype is particularly relevant due to the connection between cadherin-based junctions and the cytoskeleton.

4) Figure 5G showing a basolateral expansion phenotype that is specific of Dlg is not completely clear. This phenotype should be quantified or at least better represented to clarify if the authors refer to basolateral expansion in relation to control (basal seems larger in control) or if there are also differences to LET-413 depletion. In addition, it could be better reasoned how and if this phenotype is linked to the idea that LET-413 has additional activities independently of Dlg.

5) Since the additional activities of LET-413 could be simply explained by the weaker efficiency of depletion, the authors should provide stronger arguments to support the conclusion that there are activities independent of DLG-1. Evidence such as “LET-413 depletion does not result in a complete loss of DLG-1 yet still causes a more complete block of seam cell outgrowth “ is not sufficient because it is possible that localized Dlg is not functional in this specific process without available LET-413, such that the stronger depletion of LET-413 would create stronger functional disruption of DLG-1.

6) In the final part of the manuscript the authors test the role of Rho family GTPases in seam cell extension to investigate if these proteins could be important in the promotion of cell protrusion controlled by LET-413. Overexpression of active forms of Rac and CDC42 are well known to promote cytoskeleton reorganization and so inducing their ectopic activation would be anticipated to disrupt cytoskeleton organization regardless of a relevant function in normal conditions. Thus, ectopic overexpression experiments are insufficient to conclude about a fundamental role in regulating protrusion formation at the leading edge of extending seam cells. In fact, RNAi and expression of dominant-negative versions provide evidence that neither inactivation of Rac nor CDC42 produce any relevant defect (Fig. 6C and 6E). Although this is discussed as resulting of possible redundancy, the manuscript still lacks data to evaluate redundancy or the efficiency of inactivation. Moreover, there is no data supporting a genetic or molecular link between LET-413 and activity of these GTPases. The authors may consider testing for phenotypic interaction (e.g. co- depletion Dlg and CDC42/Rac inactivation could bring the function of CDC42/Rac as the additional activities that are responsible for the weakened phenotype of Dlg).

Given the current data, the manuscript should be considerably revised to tone down any conclusions related to the function of GTPases and its relation with the identified role for LET-413. For instance, the available data is not compatible with “an important role for the CED-10 Rac and CDC-42 GTPase proteins in reorganizing the actin cytoskeleton and driving anterior–posterior directed outgrowth of the seam cells. “; “Finally, we demonstrate that the Rho-family GTPases CED-10 Rac and CDC-42 can regulate seam cell outgrowth and may also function downstream of LET-413”; “Our data build upon these observations and support an important role for the regulation of actin dynamics by CED-10 Rac and CDC-42 in seam cell extension.”

7) The authors state in the abstract (lines 25 to 27, similar sentence in introduction in lines 88 to 90 ): “We show that the role of LET-413 in seam cell outgrowth is mediated at least in part by the junctional component DLG-1 discs large, which appears to restrict protrusive activity to the apical domain.” These sentences could be misleading as the word “restrict” suggests that DLG-1 blocks protrusive activityin all other regions of the membrane to ensure that it is limited to the apical domain. If my understanding is correct, the manuscript did not show sufficient evidence of the presence ectopic protrusive activity in the basolateral domain upon DLG-1 depletion to justify this conclusion.

MINOR ISSUES

1) Related to Major issue 1: the authors could reinforce the idea that loss of LET-413 does not disrupt apicobasal polarization by showing that junctions were not mispositioned along the apical-basal axis with imaging for HMR-1 across the lateral plane such as done for PAR6:GFP in Fig. 4C.

2) Discussion line 492: Although the authors did a great work at discussing the phenotypic difference between seam-specific RNAi from a previous study and induced protein depletion, it may be also relevant to discuss whether these differences could be related to the distinct timing of effective protein depletion.

3) Discussion line 528-531: It may be worth to consider that the distinct hierarchy defined in these cases could be related to the specific subcellular location that is being accessed in each Drosophila tissue. The Scrib module is purely located at the junctional level (septate junction) in the Drosophila midgut (Chen et al., Plos Biol) where Scrib is upstream of Dlg. In contrast, the studies in the follicular epithelium (Khoury and Bilder, PNAS, 2020; Ventura et al., Development, 2020) analyzed the basolateral localization of Scrib and show that this is indeed affected in dlg mutants. I should also note that earlier studies in a different Drosophila tissue, embryonic epithelium (Bilder et al, Science 2000), also reported Dlg upstream of Scrib.

Reviewer #3: Summary

This manuscript from the Boxem group extends their work on tissue-specific proteomics using degron technology to examine the role of a key, conserved polarity protein, LET-413/Scrib and its interactors, including DLG-1/Discs large, in postembryonic differentiation and morphogenesis in the embryonic epidermis (hypodermis) in C. elegans.

The role of LET-413 and DLG-1 in apicobasal polarity and junctional integrity was established long ago by papers from the Labouesse, Bossinger, and Hardin groups. However, while a bit of structure-function on LET-413 was subsequently performed, little has been done to examine the role of this polarity cassette in postembryonic development, which is the subject of this paper. As such this paper could be very appropriate for PLoS Genetics and should be of interest to its readers.

Interesting findings

• A basic finding of this work is perhaps not surprising: LET-413 is required in the epidermal epithelium for growth, viability, and junctional stability/maintenance. What is more surprisingly (but consistent with previous reports) is that LET-413 is not essential in the larval intestine.

• The additional, novel finding here is that LET-413 and DLG-1 play a role in the extension of seam cells (lateral epidermal cells). These cells are interesting for several reasons: they act as a stem population, show polarized division patterns, and they undergo interesting patterns of segregation from the surrounding epidermal syncytium.

• Epidermal depletion of LET-413 caused severe defects in the localization pattern of both HMR-1 and DLG-1. HMR-1 puncta became sparser, while DLG-1 was no longer localized in a continuous belt and instead localized to discontinuous short stretches.

• The authors suggest that the DLG-1/AJM-1 junctional complex is involved in spatially restricting the protrusive activity to the apical domain. This is an interesting idea, since this would be the first such evidence for such a role for DLG-1 that I am aware of. The effects of DLG-1 depletion suggest that there is an effect on basolateral localization of seam cells extensions.

Conclusions

While the basic AID methodology and associated results are interesting, there are several questions that need to be resolved regarding how LET-413 and DLG-1 act in seam cell before this work is suitable for PLoS Genetics.

Major comments

• Based on analysis of constructs designed to perturb Rho family GTPase signaling, the authors suggest that the extension process is a migratory event in which LET-413/DLG-1 restricts "protrusions" to the apical domain. This is an interesting and reasonable idea, if the actin-containing extensions are in fact protrusive. However, it seems unclear if this is the case. The authors state that "In control animals, the seam cells showed signs of active membrane dynamics immediately following cell division, displaying small filopodia-like protrusions around the cell body. The apical domains of the seam cells then formed larger lamellipodium-like protrusive fronts directed towards the adjacent seam cells", whereas these seem to disappear in the let-413-depleted larvae. Showing that these "protrusive fronts" really are protrusive, rather than relatively quiescent subapical cortex, seems important for the interpretation of the entire story here. Can the authors perform some live imaging to ascertain this?

• This issue is exacerbated by the refractory nature of these "protrusions", as the authors state: "We were not able to block formation of seam cell protrusions through mutation, RNAi, or expression of dominant negative variants of Rac family members and cdc-42." While I am agnostic on whether these are indeed "protrusions", a skeptic might say this is strong presumptive evidence that they are not. Making a bit of headway on the function of these extensions seems important. Most would worry about the CA expression results here is there are not complementary effects using loss-of-function or dominant negatives.

• Why did the authors not use ZF1 or AID technology to deplete Rho family GTPases, given that this approach is used so much in the paper? Did that also lead to no phenotype?

• Since LET-413 blocks all protrusion formation it is hard to know if the main activity of LET-413 is as an upstream regulator of DLG-1 or instead it acts through other mechanisms. The authors admit this, yet suggest that LET-413 might be upstream of CDC-42: "these results are consistent with LET-413 acting upstream to regulate CDC-42 activity, this cannot be concluded from these data as no protrusions are formed when LET-413 is depleted." I think this is correct. Making additional headway on this points would improve this paper, but seems less important than strengthening the story related to protrusions.

Other comments/questions

•I wish to strongly commend the authors for their honest interpretations and caveats throughout much of this paper. They were a pleasure to read.

•The P cell phenotype is interesting. Can the authors say anything more about this phenotype?

•What is interesting here is that the apicobasal position of the puncta is maintained, which is different from the effects of LET-413 deletion in some other tissues (e.g., embryonic epidermis and intestine). Do the authors have a suggestion as to the biological difference? Is it the presence of the cuticle?

**Have all data underlying the figures and results presented in the manuscript been provided?**

Reviewer #1: Yes

Reviewer #2: Yes

Reviewer #3: Yes

PLOS authors have the option to publish the peer review history of their article (what does this mean?). If published, this will include your full peer review and any attached files.

Reviewer #1: No

Reviewer #2: No

Reviewer #3: No

---

## [Decision Letter · Decision Letter 1]

4 Oct 2021

Dear Dr Boxem,

We are pleased to inform you that your manuscript entitled "Caenorhabditis elegans LET-413 Scribble is essential in the epidermis for growth, viability, and directional outgrowth of epithelial seam cells" has been editorially accepted for publication in PLOS Genetics. Congratulations!

Yours sincerely,

Jeremy Nance

Associate Editor

PLOS Genetics

Gregory P. Copenhaver

Editor-in-Chief

PLOS Genetics

Comments from the reviewers (if applicable):

Reviewer's Responses to Questions

**Comments to the Authors:**

Reviewer #1: The authors have done a nice job with the revisions and have satisfactorily address my comments. I also like the new experiments with Cadherin and Rho GTPases, which were suggested by the other reviewers. I have no further reservations and congratulate the authors on a nice paper.

Reviewer #2: I must praise the authors for the excellent work performed during revision. My previous concerns have been fully addressed and I feel the manuscript benefits from the new organisation as it brings the emphasis to experiments that disentangle of the role of the two junctional complexes (DLG- 1/AJM-1 and cadherin-catenin) in seam cell extension.

The current conclusions are fully supported, and any potential caveats are very well discussed. I recommend publication of the manuscript in the current version.

Reviewer #3: This revision addresses my concerns about overinterpretation, and focuses on what I believe are the key results from this work. The authors have done a nice job of addressing reviewer concerns.

**Have all data underlying the figures and results presented in the manuscript been provided?**

Reviewer #1: Yes

Reviewer #2: Yes

Reviewer #3: Yes

PLOS authors have the option to publish the peer review history of their article (what does this mean?). If published, this will include your full peer review and any attached files.

Reviewer #1: No

Reviewer #2: **Yes: **Eurico Morais-de-Sá

Reviewer #3: No

**Data Deposition**

http://datadryad.org/submit?journalID=pgenetics&manu=PGENETICS-D-21-00499R1

**Press Queries**

---

## [Editor Report · Acceptance letter]

15 Oct 2021

PGENETICS-D-21-00499R1 

Caenorhabditis elegans LET-413 Scribble is essential in the epidermis for growth, viability, and directional outgrowth of epithelial seam cells 

Dear Dr Boxem, 

We are pleased to inform you that your manuscript entitled "Caenorhabditis elegans LET-413 Scribble is essential in the epidermis for growth, viability, and directional outgrowth of epithelial seam cells" has been formally accepted for publication in PLOS Genetics! Your manuscript is now with our production department and you will be notified of the publication date in due course.

With kind regards,

Zsofia Freund

PLOS Genetics

On behalf of:
